# Traceable Black-Box Watermarks For Federated Learning

**Jiahao Xu**[1,2*]    **Rui Hu**[1]    **Olivera Kotevska**[2]    **Zikai Zhang**[1]
[1] University of Nevada, Reno    [2] Oak Ridge National Laboratory
{jiahaox, ruihu, zikaiz}@unr.edu, kotevskao@ornl.gov

## Abstract

Due to the distributed nature of Federated Learning (FL) systems, each local client has access to the global model, which poses a critical risk of model leakage. Existing works have explored injecting watermarks into local models to enable intellectual property protection. However, these methods either focus on non-traceable watermarks or traceable but white-box watermarks. We identify a gap in the literature regarding the formal definition of traceable black-box watermarking and the formulation of the problem of injecting such watermarks into FL systems. In this work, we first formalize the problem of injecting traceable black-box watermarks into FL. Based on the problem, we propose a novel server-side watermarking method, **TraMark**, which creates a traceable watermarked model for each client, enabling verification of model leakage in black-box settings. To achieve this, **TraMark** partitions the model parameter space into two distinct regions: the main task region and the watermarking region. Subsequently, a personalized global model is constructed for each client by aggregating only the main task region while preserving the watermarking region. Each model then learns a unique watermark exclusively within the watermarking region using a distinct watermark dataset before being sent back to the local client. Extensive results across various FL systems demonstrate that **TraMark** ensures the traceability of all watermarked models while preserving their main task performance. The code is available at https://github.com/JiiahaoXU/TraMark.

## 1 Introduction

Federated Learning (FL) is a promising training paradigm that enables collaborative model training across distributed local clients while ensuring that private data remains on local devices (McMahan et al., 2017). Instead of sharing raw data, clients train local models independently and periodically send updates to a central server, which aggregates them into a global model. This privacy-preserving benefit has led to FL's widespread adoption in various fields, including healthcare (Nguyen et al., 2022), finance (Long et al., 2020), and remote sensing (Liu et al., 2020), where local data privacy is a critical concern. However, sharing the global model with all participants introduces risks of model leakage. Specifically, malicious clients may exploit their access by duplicating and illegally distributing the model (Li et al., 2022a). Such misconduct undermines the integrity of the FL system and compromises the collective interests of all participants. Consequently, protecting intellectual property (IP) rights of FL-trained models and detecting copyright infringement have become critical challenges in FL (Xue et al., 2022).

Protecting the IP of FL-trained models requires mechanisms to verify rightful ownership if a model is unlawfully distributed (Shao et al., 2024) (i.e., proving that the model originated from the FL system). To address this, researchers have proposed embedding watermarks into the global model to enable ownership verification. Existing approaches primarily fall into two categories: parameter-based (Uchida et al., 2017) and backdoor-based (Tekgul et al., 2021; Adi et al., 2018) watermarking techniques. Parameter-based methods embed signatures (e.g., bit strings) within the model's parameters as a secret key. During verification, the verifier extracts this key from the suspect model and applies a cryptographic function with the corresponding public key to validate the model's ownership.

---

*Partially done during the internship at the Oak Ridge National Laboratory.

However, this process requires white-box access to the model parameters, which is often infeasible in practice, particularly when the suspect model is only accessible in a black-box setting (e.g., via an API). To overcome this limitation, backdoor-based watermarking leverages backdoor injection techniques to ensure that the model learns a specific trigger. A watermarked model outputs predefined responses when presented with inputs containing the trigger. Unlike parameter-based methods, this verification process does not require access to the model parameters, making backdoor-based watermarking a more practical solution.

Beyond ownership verification, the verifier also needs to trace the source of model leakage, identifying which client was responsible for the unauthorized distribution. Recent studies have explored methods for ensuring the traceability of watermarked models in FL (Shao et al., 2024; Yu et al., 2023; Xu et al., 2024; Nie & Lu, 2024a). For instance, FedTracker (Shao et al., 2024) embeds unique bit strings into each client's model and identifies the leaker by measuring bit string similarities. However, this approach requires white-box access to the suspect model, which limits its practicality. Another approach, FedCRMW (Nie & Lu, 2024a), introduces a black-box watermarking mechanism that injects unique watermarks into each model shared with the client by mixing clients' local datasets with multiple types of triggers. The model leaker is then identified based on the models' predictions on these watermarked datasets. However, this method modifies the local training protocol and requires access to clients' local data, making it vulnerable to tampering by malicious clients. A recent method, MFL-Owner (Gai et al., 2025), targets multi-modal FL by leveraging each client's visual and language encoders to construct an orthogonal transformation on the client's trigger set, which serves as a watermark. However, this approach suffers from poor generalizability and fails to scale effectively to broader FL systems. Despite notable empirical progress, existing literature still lacks a formal problem definition and formulation of black-box watermarking for ownership verification and traceability in FL.

In this work, we first formalize the problem of traceable black-box watermarking injection in FL systems. Building on this, we propose **TraMark**, which creates a personalized, traceable watermarked model for each client, enabling verification of model leakage in ***black-box*** settings. Specifically, to ensure traceability, the server partitions the model's parameter space into two regions: the *main task region*, responsible for learning the primary FL task, and the *watermarking region*, designated for embedding a watermark. The server then generates personalized global models for each client via *masked aggregation*. Each model is subsequently injected with a distinct watermark exclusively in the watermarking region using a dedicated watermark dataset. This process ensures that every client receives a personalized global model that integrates aggregated knowledge from other clients while embedding a unique watermark for model leakage verification. We summarize the contributions of our paper as follows.

- To the best of our knowledge, this is the first work to formally formulate the problem of traceable black-box watermarking in FL systems. Based on this, we propose **TraMark**, a novel watermarking method that can be seamlessly integrated into existing FL systems.

- **TraMark** is designed to inject unique watermarks into models shared with clients while preventing watermark collisions in FL, enabling the identification of model leakers in black-box settings.

- We demonstrate the effectiveness of **TraMark** through extensive experiments across various FL settings. Results show that **TraMark** ensures clients receive traceable models while maintaining main task performance, with only a slight average drop of $0.54\%$. Additionally, we conduct a detailed hyperparameter analysis of **TraMark** to evaluate the impact of each configuration on both main task performance and leakage verification.

## 2    BACKGROUND AND SYSTEM SETTINGS

**Federated Learning.** A typical FL system consists of a central server and a set of $n$ local clients, which collaboratively train a shared model $\theta \in \mathbb{R}^d$. The FL problem is generally formulated as: $\min_\theta (1/n) \sum_{i=1}^n F_i(\theta; \mathcal{D}_i^l)$, where $F_i(\cdot)$ represents the local learning objective of client $i$, and $\mathcal{D}_i^l$ is its local dataset. For instance, for a classification task, client $i$'s local objective can be expressed as: $F_i(\theta; \mathcal{D}_i^l) := \mathbb{E}_{(z,y) \in \mathcal{D}_i^l} \mathcal{L}(\theta; z, y)$, where $\mathcal{L}(\cdot)$ is the loss function, and $(z, y)$ represents a datapoint sampled from $\mathcal{D}_i^l$. A classic method to solve the FL problem is Federated Averaging (FedAvg) (McMahan et al., 2017). Specifically, in each training round $t$, the server broadcasts the

current global model $\theta^t$ to each client $i \in [n]$. Upon receiving $\theta^t$, client $i$ performs $\tau_l$ iterations of local training on it using $\mathcal{D}_i^l$, resulting in an updated local model $\theta_i^{t,\tau}$. The client then computes and sends the model update $\Delta_i^t = \theta_i^{t,\tau} - \theta^t$ back to the server. The server aggregates updates from all clients and refines the global model as: $\theta^{t+1} = \theta^t + (1/n) \sum_{i=1}^{n} \Delta_i^t$. This process repeats until the global model converges.

**Attack Model.** In this work, we consider a widely used attack model where all clients in an FL system are potential model leakers (Shao et al., 2024; Tekgul et al., 2021). Specifically, we assume these clients will follow a predefined local training protocol to complete the FL task. However, they may illegally distribute their local models for personal profit.

**Defense Model.** In this work, we consider the standard FL setup (e.g., FedAvg) where the server directly receives and aggregates local model updates but has no access to any private local data. The server is considered always reliable and equipped with sufficient computational resources. We assume that the server acts as the defender, responsible for injecting traceable watermarks into the FL system. Moreover, the server aims to keep the watermark injection process confidential from all local clients. Additionally, the server also acts as the verifier: if a model is deemed suspicious, it initiates a verification process to determine whether the model originates from the FL system and to identify the responsible model leaker.

## 3 PROBLEM FORMULATION

**Black-box Watermarking.** A black-box watermark is a practical watermarking solution that enables verification without access to model parameters, making it more suitable for real-world deployment. A common black-box watermarking approach leverages backdoor injection (Adi et al., 2018), where a watermark dataset $\mathcal{D}^w$ (as defined in Definition 1) containing triggers is used to train the model to produce a predefined output when presented with the triggers.

**Definition 1** (Watermark Dataset). *A watermark dataset $\mathcal{D}^w$ is a designated set of trigger-output pairs used to embed a watermark into a model. Formally,*

$$\mathcal{D}^w = \{(x, \phi(x)) \mid x \in \mathcal{X}^w\},$$

*where $\phi(x)$ is the unique predefined output distribution assigned to each trigger $x$ in trigger set $\mathcal{X}^w$.*

With the watermark dataset, we define the black-box watermark as follows.

**Definition 2** (Black-box Watermark). *A valid black-box watermark $\delta$ is a carefully crafted perturbation learned from the watermark dataset $\mathcal{D}^w$. It is applied to a model $\theta$ to obtain the watermarked model $\theta' = \theta + \delta$, which produces outputs following the predefined distribution $\phi(x)$ when evaluated on $\mathcal{D}^w$. Formally, the model $\theta'$ is considered watermarked with $\delta$ if:*

$$\mathbf{y}(\theta'; x) \sim \phi(x), \quad \forall (x, \phi(x)) \in \mathcal{D}^w,$$

*where $\mathbf{y}(\theta'; x)$ is the output probability distribution of the watermarked model given a trigger $x$.*

If a black-box watermark is successfully embedded, one can verify whether a suspicious model originates from the system by testing its outputs on triggers from $\mathcal{D}^w$. For example, in classification tasks, verification is typically performed by evaluating the *prediction accuracy* of the suspicious model $\theta'$ on $\mathcal{D}^w$. Specifically, if the model's prediction accuracy $\sum_{x \in \mathcal{D}^w} \mathbf{1}[\arg\max \mathbf{y}(\theta'; x) = \arg\max \phi(x)]/|\mathcal{D}^w|$ exceeds a predefined threshold $\nu$, this indicates that the suspicious model contains the watermark $\delta$, thereby verifying its ownership (Tekgul et al., 2021; Shao et al., 2024; Liu et al., 2021; Nie & Lu, 2024b).

**Traceability.** However, ownership verification alone is insufficient for detecting model leaking of an FL system. While ownership verification confirms whether a model originates from the system, it does not identify which client leaked it. The ability to pinpoint the source of leakage is known as *traceability*. To achieve traceability, each watermarked model should carry a distinct watermark, ensuring that every client receives a unique identifier embedded in their model. More formally, for any two watermarked models $\theta_i'$ and $\theta_j'$, their outputs should be as different as possible when evaluated on the same watermark dataset. If their outputs are too similar, a watermark collision (as defined in Definition 3) occurs, which can compromise traceability.

**Definition 3** (Watermark Collision). *A watermark $\delta_i$ learned from a watermark dataset $\mathcal{D}_i^w$ is said to collide with another watermark $\delta_j$ learned from $\mathcal{D}_j^w$ if their corresponding watermarked models, $\theta_i' = \theta + \delta_i$ and $\theta_j' = \theta + \delta_j$, produce highly similar outputs on watermark dataset $\mathcal{D}_i^w$ or $\mathcal{D}_j^w$. Formally, a collision occurs if:*

$$\mathbb{E}_x\big[\mathtt{Div}\big(\mathbf{y}(\theta_i';x), \mathbf{y}(\theta_j';x)\big)\big] \leq \sigma,$$

*for $x \in \mathcal{D}_i^w$ or $x \in \mathcal{D}_j^w$, where $\mathtt{Div}(\cdot)$ is a divergence measurement function (e.g., KL divergence), and $\sigma$ is a predefined collision threshold.*

**Remark 1.** *Watermark collision poses a significant challenge in ensuring the traceability of watermarked models. Since $\delta_i$ and $\delta_j$ are learned from $\mathcal{D}_i^w$ and $\mathcal{D}_j^w$, respectively, the distinctiveness of these watermark datasets plays a crucial role in preventing collisions. Specifically, if $\mathcal{D}_i^w$ and $\mathcal{D}_j^w$ are too similar, the resulting $\delta_i$ and $\delta_j$ will also be similar, increasing the risk of collision. Therefore, it is essential to ensure an intrinsic difference between $\mathcal{D}_i^w$ and $\mathcal{D}_j^w$. We discuss strategies for constructing distinct watermark datasets to mitigate collisions in Section 4.3.*

With Definition 1–3, we formally define the traceability of watermarked models as follows.

**Definition 4** (Traceability of Watermarked Models). *Given $n$ watermarked models $\{\theta_1', \theta_2', \ldots, \theta_n'\}$, where each model is derived as $\theta_i' = \theta + \delta_i$, with a successfully embedded watermark $\delta_i$, such that $\mathbf{y}(\theta_i';x) \sim \phi(x), \; \forall(x, \phi(x)) \in \mathcal{D}_i^w$. The traceability property ensures that different watermarked models produce distinguishable outputs on their respective watermark datasets. Formally, if for any watermarked model $\theta_i', \; i \in [n]$, the following holds:*

$$\mathbb{E}_{x\in\mathcal{D}_i^w}\big[\mathtt{Div}\big(\mathbf{y}(\theta_i';x), \mathbf{y}(\theta_j';x)\big)\big] > \sigma, \quad \forall j \in [n], j \neq i.$$

*then the traceability of these models is ensured.*

Intuitively, if each watermark in the system remains distinct and does not collide with any other watermark, then all watermarked models in the system are considered traceable.

**Problem Formulation.** We define the problem of injecting traceable black-box watermarks in FL as in Problem 1.

**Problem 1** (Traceable Black-box Watermarking). *Consider an FL system with $n$ clients collaboratively training a global model $\theta$ under the coordination of the server. For watermark injection, the server prepares $n$ distinct watermark datasets $\{\mathcal{D}_i^w\}_{i=1}^n$ to be used to inject watermarks into the global models for every client. The overall goal is to optimize both the main task learning objective and the watermarking objective while ensuring that the watermarked models remain traceable. This is formulated as follows:*

$$\min_{\theta,\{\delta_i\}_{i=1}^n} \underbrace{\frac{1}{n}\sum_{i=1}^n F_i(\theta;\mathcal{D}_i^l)}_{\text{Main Task}} + \underbrace{\frac{1}{n}\sum_{i=1}^n L_i(\theta+\delta_i;\mathcal{D}_i^w)}_{\text{Watermarking Task}},$$

$$s.t.\ \mathbb{E}_{x\in\mathcal{D}_i^w}\big[\mathtt{Div}(\mathbf{y}(\theta+\delta_i;x), \mathbf{y}(\theta+\delta_j;x))\big] > \sigma, \quad \forall i,j \in [n],\ i \neq j,$$

*where $\delta_i$ denotes the traceable black-box watermark for the model $\theta_i$ shared with client $i$. The function $L_i(\cdot)$ represents the watermarking objective for $\delta_i$, defined as: $L_i(\theta+\delta_i;\mathcal{D}_i^w) := \mathbb{E}_{(x,\phi(x))\in\mathcal{D}_i^w}\mathcal{L}(\theta+\delta_i;x,\phi(x))$.*

**Remark 2.** *We derive the following key insights, which motivate the design of our method:*

*1) A straightforward solution for Problem 1 is to offload each watermark dataset $\mathcal{D}_i^w$ to client $i$, allowing clients to mix $\mathcal{D}_i^w$ with their main task dataset $\mathcal{D}_i^l$ during local training to solve both objectives simultaneously, as proposed in (Nie & Lu, 2024a;b; Liu et al., 2021; Xu et al., 2024; Wu et al., 2022; Li et al., 2022a). However, this method is highly vulnerable to malicious clients who may simply discard $\mathcal{D}_i^w$, leading to the absence of watermarks in their models. Furthermore, if malicious clients are aware of the watermarking process, they could intentionally tamper with it, undermining its effectiveness. To mitigate these risks, it is preferable to decompose Problem 1, leaving the main task to local clients while performing watermark injection solely on the server.*

*2) A critical challenge in watermark injection is the risk of watermark collisions due to model averaging during aggregation. Specifically, even if the server successfully injects distinct watermarks*

*into the models before sending them to clients for local training, these watermarks will be fused during model aggregation in the next training round if parameters from all clients are simply averaged, as in FedAvg. To address this issue, a specialized mechanism is required to prevent watermark entanglement during aggregation.*

*3) The constraint in Problem 1 suggests that to avoid collisions, the server should maximize $\|\delta_i - \delta_j\|_2^2$. However, if $\delta_i$ and $\delta_j$ become too divergent, this may impair the main task performance of the watermarked models. Additionally, since the server lacks access to clients' local datasets, directly solving the watermarking objective $L(\cdot)$ could also lead to significant degradation in the performance of the main task. Thus, a specialized learning strategy is required to ensure that watermark injection does not compromise the model's main task performance.*

# 4 INJECTING TRACEABLE BLACK-BOX WATERMARKS

Based on the insights in Remark 2, we propose a novel method called **TraMark** detailed in Algorithm 1, which can be easily integrated into existing FedAvg frameworks to solve Problem 1. We give the complete process of FedAvg with **TraMark** in Algorithm 2 in the Appendix A.3.

## 4.1 CONSTRAINING WATERMARKING REGION

Existing watermarking approaches either retrain the global model directly on the watermark dataset (Tekgul et al., 2021; Shao et al., 2024) or require local clients to collaboratively inject watermarks (Li et al., 2022a; Liu et al., 2021; Nie & Lu, 2024b). However, these methods cause watermark-related perturbations to spread across the entire parameter space, leading to two key issues. First, the dispersed watermark perturbations may significantly degrade the main task performance. Second, even if each client's model embeds a unique watermark, model aggregation in the next round fuses these watermarks, causing collisions that compromise traceability. To mitigate the impact on main task performance and ensure traceability, **TraMark** restricts watermarking to a small subset of the model's parameter space. Only this designated watermarking region carries the watermark, and its parameters are excluded from model aggregation, preserving distinct watermarks for each client in the next training round. Specifically, in

---

**Algorithm 1: TraMark**

**Input** : The global models $\{\theta_i\}_{i=1}^n$, a set of model updates $\{\Delta_i\}_{i=1}^n$, watermark datasets $\{\mathcal{D}_i^w\}_{i=1}^n$, main task mask $\mathbf{M}_m$, watermarking mask $\mathbf{M}_w$, watermarking learning rate $\eta_w$, and watermarking iteration $\tau_w$.

**Output:** A set of watermarked models $\{\theta_i'\}_{i=1}^n$.

// **Watermark injection**

1 **for** $i \in [n]$ **do**

    // Masked aggregation

2     $\tilde{\theta}_i \leftarrow \theta_i + \mathbf{M}_m \odot \frac{1}{n}\sum_{i=1}^n \Delta_i + \mathbf{M}_w \odot \Delta_i$

    // Watermarking

3     $\tilde{\theta}_i^0 \leftarrow \tilde{\theta}_i$

4     **for** $s = 0$ **to** $\tau_w - 1$ **do**

5         $g_i^s \leftarrow \nabla_{\tilde{\theta}_i^s} \mathcal{L}(\tilde{\theta}_i^s; \mathcal{D}_i^w)$

6         $\tilde{\theta}_i^{s+1} \leftarrow \tilde{\theta}_i^s - \eta_w g_i^s \odot \mathbf{M}_w$

7     **end**

8     $\theta_i' \leftarrow \tilde{\theta}_i^{\tau_w}$

9 **end**

10 **Return** $\{\theta_i'\}_{i=1}^n$

---

**TraMark**, given a model $\theta \in \mathbb{R}^d$, it partitions the whole parameter space into *watermarking region* and *main task region* with a partition ratio $k \in [0, 1)$, resulting in two *complementary* binary masks:

- The **watermarking mask** $\mathbf{M}_w \in \{0, 1\}^d$, where $[\mathbf{M}_w]_j = 1$ means that the $j$-th parameter in $\theta$ is used for *watermarking task* and $\texttt{sum}(\mathbf{M}_w) = k \times d$.
- The **main task mask** $\mathbf{M}_m \in \{0, 1\}^d$, where $[\mathbf{M}_m]_p = 1$ means that the $p$-th parameter in $\theta$ is used for *main task* and $\texttt{sum}(\mathbf{M}_m) = (1 - k) \times d$.

These two complementary masks (i.e., $\mathbf{M}_w + \mathbf{M}_m = \mathbf{1}^d$) ensure that all model parameters in the model $\theta$ are *fully* partitioned into the watermarking and main task regions. Moreover, once determined, the masks remain unchanged throughout the entire watermarking process. We will discuss how to get these two masks later in Section 4.4.

## 4.2 MASKED AGGREGATION & WATERMARK INJECTION

**Masked Aggregation.** With the constrained watermarking region, **TraMark** leverages a novel *masked aggregation* method to avoid watermark collision. Specifically, instead of applying a naive

aggregation approach (e.g., FedAvg), **TraMark** aggregates parameters only in the main task region and prevents the watermarking region from parameter fusion. In detail, in each training round, given the model updates $\{\Delta_i\}_{i=1}^n$, the server aggregates them to generate the personalized global model for each client individually via $\tilde{\theta}_i = \theta_i + \mathbf{M}_m \odot \frac{1}{n} \sum_{i=1}^n \Delta_i + \mathbf{M}_w \odot \Delta_i, \ \forall i \in [n]$ (Line 2 in Algorithm 1). Here, the second term $\mathbf{M}_m \odot \frac{1}{n} \sum_{i=1}^n \Delta_i$ averages the model updates in the main task region, and the third term $\mathbf{M}_w \odot \Delta_i$ preserves the parameters in the watermarking region for the model update of client $i$. In this case, the server generates a personalized global model for each client, which always contains a *distinct watermark* for the client while still benefiting from the aggregated model updates for the main task.

**Watermark Injection.** For each personalized global model $\tilde{\theta}_i, \forall i \in [n]$, its watermark is injected by training $\tilde{\theta}_i$ on the corresponding distinct watermark dataset $\mathcal{D}_i^w$ for $\tau_w$ iterations. In this process, only the watermarking region is updated, ensuring that knowledge from the watermark dataset does not spread to the main task region (Line 4–7). Technically, in each step of local training, the mini-batch gradient $g_i^s$ is multiplied by $\mathbf{M}_w$ (Line 5–6), zeroing out the gradients for the main task region to avoid the impact of the watermark on the main task. After watermark injection, the server obtains the watermarked global models, which will be sent to the clients to perform their main tasks of the next training round or deployment (Line 10). Since the clients' local training protocol for the main task remains unchanged, model updates continue across the entire parameter space. However, this introduces a potential risk: the embedded watermarks may gradually fade over time. To this end, **TraMark** can be applied at every training round to continuously preserve the watermark, as advised by prior works (Tekgul et al., 2021; Shao et al., 2024; Nie & Lu, 2024b; Li et al., 2022a).

### 4.3 Distinct Watermark Dataset

As noted in Remark 1, ensuring sufficient dissimilarity between watermark datasets is crucial for learning distinct watermarks and preventing collisions. To achieve this, in the **TraMark**, the server assigns each global model a unique watermark dataset. Specifically, the watermark datasets designed for different models should differ from each other in both their triggers and their output distributions. Let $\mathcal{D}_i^w = \{(x, \phi_i(x)) \mid x \in \mathcal{X}_i^w\}$ denote the watermark dataset for personalized global model $\tilde{\theta}_i$, where $\mathcal{X}_i^w \cap \mathcal{X}_j^w = \emptyset$ for any $i \neq j$, $\forall i, j \in [n]$. Furthermore, each client is assigned a unique output distribution $\phi_i(x)$, guaranteeing that $\phi_i(x) \neq \phi_j(x)$. For trigger selection, existing methods have explored various approaches, including randomly generated patterns (Tekgul et al., 2021; Shao et al., 2024), adversarially perturbed samples (Li et al., 2022a), and samples embedded with backdoor triggers (Liu et al., 2021). In our case, to ensure each client receives a maximally distinct trigger, we select out-of-distribution samples absent from the main task dataset. This guarantees that the learned watermark remains independent of the main task. For example, in a classifier trained for traffic sign recognition, per-label samples from the MNIST dataset serve as effective triggers for different clients. Assigning a distinct watermark dataset to each personalized global model ensures that a watermarked model responds only to triggers from its own dataset, mapping them to the predefined output. When exposed to triggers from other watermark datasets, it produces random guesses, effectively minimizing the risk of collisions. This distinct watermark dataset assignment may limit the scalability of **TraMark**, particularly in learning tasks with only a few labels. We discuss possible solutions to this limitation in Appendix A.9.

### 4.4 Selection of Watermarking Region

Recall from Remark 2 that maximizing $\|\delta_i - \delta_j\|_2^2$ is crucial for avoiding collisions. However, excessive divergence between $\delta_i$ and $\delta_j$ may negatively impact main task performance. Given that $\|\delta_i - \delta_j\|_2^2 = \|\mathbf{M}_m \odot (\delta_i - \delta_j)\|_2^2 + \|\mathbf{M}_w \odot (\delta_i - \delta_j)\|_2^2$, where $\mathbf{M}_m = \mathbf{1}^d - \mathbf{M}_w$ and the watermarking process is confined to the watermarking region, the objective simplifies to maximizing $\|\mathbf{M}_w \odot (\delta_i - \delta_j)\|_2^2$. This ensures that watermark injection and collision avoidance should not affect parameters in the main task region. Consequently, if $\mathbf{M}_m$ contains the most important parameters while $\mathbf{M}_w$ is assigned to unimportant ones, the primary accuracy remains largely unaffected. Typically, parameter importance is measured by magnitude (absolute value), with larger values indicating greater importance (Xu et al., 2025; Hu et al., 2023; Panda et al., 2022). However, since network parameters are randomly initialized at the start of training, their importance is not yet established. As a result, assigning parameters to regions too early may lead to suboptimal partitioning, potentially

degrading main task performance. To this end, **TraMark** introduces a *warmup training phase*, where the global model undergoes standard federated training (e.g., FedAvg) for $\alpha \times T$ rounds before watermarking. The warmup training ratio $\alpha \in [0, 1)$ determines the fraction of total training rounds allocated to this phase, ensuring the model is robust enough to the main task before starting the watermark injection. Once warmup training is complete, the server obtains the watermarking region by selecting $k \times d$ least important parameters (i.e., those that have the smallest absolute values), and the remaining parameters are assigned to the main task region. We present the detailed technical procedure of selecting the watermarking region in Algorithm 2, included in Appendix A.3.

## 5 EXPERIMENTS

### 5.1 EXPERIMENTAL SETTINGS

**General Settings.** Following (Shao et al., 2024; Zhang et al., 2024; Nie & Lu, 2024a;b), we evaluate our methods on FMNIST (Xiao et al., 2017), CIFAR-10 (Krizhevsky et al., 2009), and CIFAR-100 (Krizhevsky et al., 2009), using a CNN, AlexNet (Krizhevsky et al., 2012), and VGG-16 (Simonyan, 2014), respectively. Besides these benchmarks, we test **TraMark** on large-scale Tiny-ImageNet using ViT (Dosovitskiy et al., 2021). For all datasets, we consider both independent and identically distributed (IID) data and non-IID data scenarios. To simulate non-IID cases, we use the Dirichlet distribution (Minka, 2000) with a default degree $\gamma = 0.5$. Following previous works (Li et al., 2022a; Xu et al., 2024; Nie & Lu, 2024a;b), we set up a cross-silo FL system with 10 clients. We also test **TraMark** on larger-scale $n = 50$ FL with client sampling in Appendix A.7. Each client performs local training with $\tau_l = 5$ iterations and a learning rate of $\eta_l = 0.01$. The training rounds for FMNIST, CIFAR-10, CIFAR-100, and Tiny-ImageNet are $50, 100, 100$, and $50$. *All experiments are repeated 3 times with different seeds, and we report the averaged results.*

**Baselines and TraMark Settings.** In all experiments, we use the MNIST (LeCun et al., 1998) dataset as the source for watermarking. Each global model is assigned a watermark dataset containing samples from a distinct MNIST label, ensuring both $\mathcal{X}_i^w \cap \mathcal{X}_j^w = \emptyset$ and $\phi_i(x) \neq \phi_j(x)$ for any two clients $i$ and $j$. Furthermore, we demonstrate that **TraMark** can accommodate various types of watermark datasets, including randomly generated patterns as proposed in (Tekgul et al., 2021), with results provided in Section 5.3. Each watermark dataset consists of 100 samples. The watermarking learning rate is set to $\eta_w = 1e^{-4}$, and the number of watermarking iterations is $\tau_w = 5$. The partition ratio $k$ is set to 1%. The warmup training ratio $\alpha$ is set to 0.5. To ensure a fair comparison, we evaluate **TraMark** against two *server-side* watermarking approaches: WAFFLE (Tekgul et al., 2021), a black-box watermarking method that does not ensure traceability, and FedTracker (Shao et al., 2024), a white-box watermarking method that ensures traceability.

**Evaluation Metrics.** We evaluate the performance of each method using two key metrics: main task accuracy (MA) and model leakage verification rate (VR). MA is measured using the main task test set. Since **TraMark** and FedTracker introduce slight variations in each local model due to watermark injection, we compute MA as the average accuracy across all local models, following (Shao et al., 2024). VR quantifies the proportion of watermarked models that are successfully attributed to their respective owners. We evaluate each watermarked model on the full test set, compute the per-label accuracy, and identify the label with the highest accuracy. If this highest-accuracy label matches the pre-assigned label of the model's owner, the model is considered successfully verified (more details are given in Appendix A.4). For FedTracker, we follow its original definition of VR, where traceability is determined based on fingerprint similarity in a white-box setting.

### 5.2 MAIN EMPIRICAL RESULTS

**Verification Interval.** We first demonstrate the verifier's confidence in identifying the leaker of suspect models embedded with watermarks injected by **TraMark**. Specifically, we compute two key metrics: (1) the test accuracy of each watermarked model on its own watermarking dataset, referred to as *verification confidence*; and (2) the average test accuracy of the model on other clients' watermarking datasets, referred to as *verification leakage*. The difference between these two metrics, termed the *verification interval*, reflects the verifier's confidence. Figure 1 illustrates the averaged verification confidence, leakage, interval, and the VR changes across training rounds on CIFAR-10 and CIFAR-100 datasets.

We observe that as training progresses, the verification interval widens due to a steady increase in verification confidence. Additionally, since **TraMark** injects watermarks only within the designated watermarking region and employs masked aggregation, the watermarked model consistently performs poorly on other clients' watermarking datasets. These factors together form a clear verification interval, which ensures reliable model leakage verification. This is supported

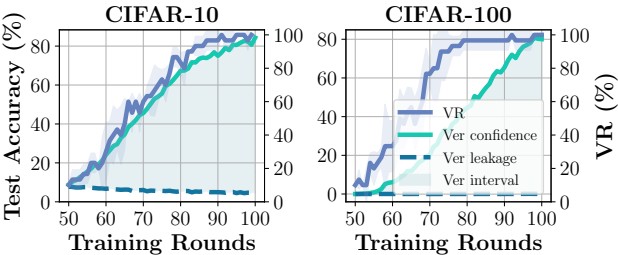

Figure 1: Verification (Ver) confidence, verification leakage, verification interval, and VR on CIFAR-10 and CIFAR-100.

by the high VR obtained by **TraMark**. Further results on the changes in the divergence of each watermarked model's output on its watermark dataset (the constraint in Problem 1) are provided in Appendix A.6.

**Main Results.** We report the MA and VR of each method on FMNIST (FM), CIFAR-10 (C-10), CIFAR-100 (C-100), and Tiny-ImageNet (Tiny) under both IID and non-IID settings in Table 1. Overall, **TraMark** effectively injects traceable watermarks into personalized global models while preserving high model performance. It consistently achieves a high VR across all datasets, maintaining an average of 99.58%. In contrast, FedTracker fails to ensure sat-

Table 1: Comprehensive comparison of MA and VR between different methods under both IID and non-IID settings (highlighted with a "N"). MAs and VRs are shown in percentages (%).

| Dataset | FedAvg | | WAFFLE | | FedTracker | | TraMark | |
|---|---|---|---|---|---|---|---|---|
| | MA | VR | MA | VR | MA | VR | MA | VR |
| FM | 92.60 | - | 92.21 | - | 89.95 | 100.00 | 91.20 | 96.67 |
| FM (N) | 91.52 | - | 91.41 | - | 67.50 | 100.00 | 91.31 | 100.00 |
| C-10 | 89.15 | - | 89.16 | - | 87.56 | 60.00 | 88.58 | 100.00 |
| C-10 (N) | 87.01 | - | 86.75 | - | 83.42 | 50.00 | 86.26 | 100.00 |
| C-100 | 61.91 | - | 61.68 | - | 61.05 | 100.00 | 61.13 | 100.00 |
| C-100 (N) | 60.19 | - | 60.04 | - | 60.12 | 90.00 | 58.95 | 100.00 |
| Tiny | 21.05 | - | 21.24 | - | 20.40 | 100.00 | 20.91 | 100.00 |
| Tiny (N) | 20.09 | - | 19.97 | - | 20.00 | 100.00 | 20.06 | 100.00 |
| **Average** | 65.44 | - | 65.31 | - | 61.25 | 87.50 | 64.90 | 99.58 |

isfactory traceability on CIFAR-10, resulting in an average VR of only 87.50%. This instability is due to the injection of key matrices into model parameters and the absence of explicit mechanisms to prevent watermark collisions after aggregation. Regarding MA, **TraMark** exhibits strong model performance, with only a 0.54% drop compared to FedAvg. While WAFFLE achieves a slightly higher average MA (0.41% above **TraMark**), it does not guarantee the traceability of watermarked models. FedTracker, on the other hand, suffers a significant 4.19% decline in MA due to the unconstrained watermarking region, which compromises model utility. In conclusion, **TraMark** successfully embeds traceable black-box watermarks while incurring minimal performance loss, making it a robust and practical watermarking solution for FL.

**Robustness to Attacks.** We evaluate the robustness of watermarked models trained by **TraMark** against pruning and fine-tuning attacks. Specifically, malicious clients may prune or fine-tune their local models to remove or reduce the effectiveness of watermarks. For the pruning attack, we test pruning ratios ranging from 30% to an extreme 99%. For fine-tuning attacks, we assume that malicious clients fine-tune their models on their local datasets for 30

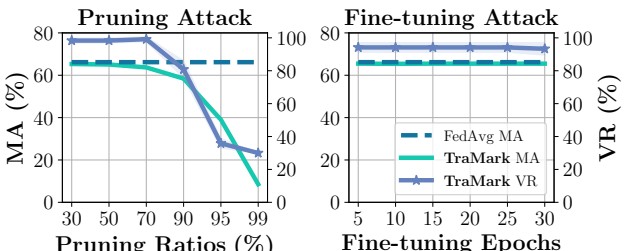

Figure 2: Averaged MA and VR results of **TraMark** under pruning and fine-tuning attacks.

epochs, using the same learning rate as in the original training process. The averaged MA and VR results of **TraMark** across four datasets are summarized in Figure 2. With moderate pruning ratios (30% to 70%), MA remains largely unaffected, while VR is also preserved. As the pruning ratio increases, MA declines rapidly, accompanied by a decrease in VR. These results demonstrate

that the parameters in the watermarking region are coupled with the main task parameters, making simple magnitude-based pruning ineffective in removing the watermarks. This coupling also contributes to stable VR across various fine-tuning epochs. Additionally, we evaluate **TraMark** against fine-tuning attacks with different learning rates, quantization attacks, and the stronger adaptive RNP attack (Li et al., 2023) in Appendix A.8. The extended results further show the robustness of **TraMark**.

## 5.3 DETAILED STUDIES OF **TraMark**

**Warmup Training Ratio** $\alpha$. **TraMark** leverages warmup training to achieve better partitioning between the main task region and the watermarking region. Figure 3 presents the averaged MA and VR of **TraMark** on four datasets under both IID and non-IID settings with varying warmup training ratios $\alpha$. Notably, **TraMark** consistently achieves satisfactory VR with $\alpha \leq 0.5$. When $\alpha = 0.7$, **TraMark** fails to inject effective watermarks

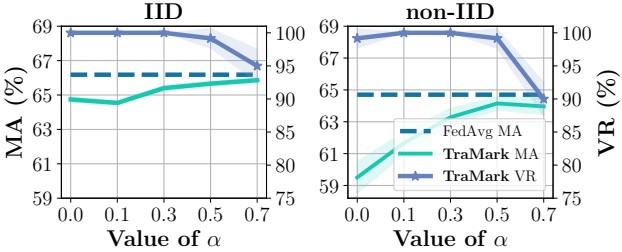

Figure 3: Impact of warmup training ratio $\alpha$ on the MA and VR of **TraMark**.

into each personalized global model *on time*, leading to degraded VR. For both settings, a larger $\alpha$ generally results in a higher MA. Specifically, under the non-IID setting, **TraMark** with the default $\alpha = 0.5$ achieves an average MA of $64.15\%$, which is only $0.55\%$ lower than FedAvg but $4.65\%$ higher than **TraMark** without warmup training. These results highlight the importance of warmup training, as it enables **TraMark** to accurately assign unimportant parameters to the watermarking region, thereby minimizing the negative impact of watermarking on main task performance in watermarked models.

**Partition Ratio** $k$. The partition ratio $k$ controls the size of the watermarking region. Intuitively, a small $k$ may hinder the watermark injection process as the watermarking region is unable to learn watermark-related information completely. We vary the partition ratio $k$ from $0.1\%$ to $5\%$ to examine its impact on the performance of **TraMark**. The averaged MA and VR results across all datasets are shown in the left sub-figure of Figure 4. As expected, a smaller $k$ leads

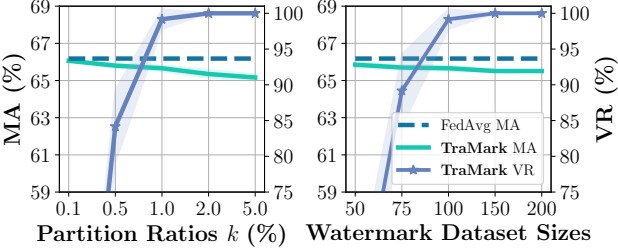

Figure 4: Impact of partition ratio $k$ and watermark dataset size.

to a significant drop in VR, while MA remains nearly unchanged. For example, compared to **TraMark** with the default setting ($k = 1.0\%$), reducing $k$ to $0.5\%$ causes VR to drop from $99.17\%$ to $84.17\%$, whereas MA shows only a slight increase from $65.66\%$ to $65.70\%$. Moreover, with an extreme value of $k = 5.0\%$, MA only drops to $65.16\%$, resulting in a gap of less than $1\%$, while achieving full VR. Therefore, selecting an appropriate $k$ requires balancing MA and VR, with $k = 1.0\%$ serving as a practical choice that ensures both reliable watermark injection and minimal performance degradation.

**Size of Watermark Dataset.** A larger watermark dataset may enhance the effectiveness of watermark injection, enabling the model to learn a more robust mapping between the watermark trigger and its intended response. To validate this, we vary the size of the watermark dataset from 50 to 200 samples. The averaged MA and VR results across all datasets are summarized in the right sub-figure of Figure 4. Similar to the effect of the partition ratio $k$, the size of the watermark dataset significantly impacts the traceability of watermarked models, while having minimal effect on MA. Specifically, when the watermark dataset contains only 50 triggers, **TraMark** achieves a suboptimal VR of $54.17\%$, despite obtaining the highest MA of $65.85\%$. However, with 100 or more samples, **TraMark** ensures successful watermark injection, with only a limited MA drop (at most

0.34%), striking a better balance between MA and VR. Therefore, choosing a sufficiently large watermark dataset, such as 100 samples or more, is essential to ensure the effectiveness of **TraMark**.

**Different Type of Watermark Dataset.** We also use the SVHN (Netzer et al., 2011) dataset and WafflePattern (Tekgul et al., 2021) as sources for watermark datasets. Since SVHN and WafflePattern contain colorful images while FMNIST is grayscale, we conduct experiments on CIFAR-10, CIFAR-100, and Tiny-ImageNet. The MA and VR results are summarized in Table 2. We observe that **TraMark** achieves highly similar results regardless of the watermark dataset used. Furthermore, we perform a T-test on each dataset to compare the results of **TraMark** using SVHN or WafflePattern against MNIST. The obtained p-values across all main task datasets exceed the commonly used significance threshold of 0.05, indicating that the differences are not statistically significant. These results demonstrate the generalization ability of **TraMark** in selecting different watermarking datasets. Furthermore, since WafflePattern utilizes synthetic noise with random patterns, it can generate an arbitrary number of unique samples. **TraMark** exploits this to assign distinct watermark datasets to every client without the need for real-world data collection.

Table 2: Generalization of **TraMark** across different watermarking datasets, including MNIST, SVHN, and WafflePattern. MAs and VRs are shown in percentages (%).

| Watermark Dataset | CIFAR-10 | | CIFAR-100 | | Tiny-ImageNet | |
|---|---|---|---|---|---|---|
| | MA | VR | MA | VR | MA | VR |
| MNIST | 88.58 | 100.0 | 61.13 | 100.0 | 20.91 | 100.0 |
| SVHN | $88.52_{p \geq 0.05}$ | 100.0 | $60.92_{p \geq 0.05}$ | 100.0 | $20.22_{p \geq 0.05}$ | 100.0 |
| WafflePattern | $88.84_{p \geq 0.05}$ | 100.0 | $61.31_{p \geq 0.05}$ | 100.0 | $20.91_{p \geq 0.05}$ | 100.0 |

**Computational Overhead.** We evaluate the server-side computational cost of the watermarking process, with results on CIFAR-10 dataset summarized in Table 3. Since **TraMark** generates personalized models to ensure traceability, the computational cost scales linearly with the number of participating clients, a characteristic shared by the state-of-the-art traceable method, FedTracker. However, the absolute overhead remains highly efficient. As shown in the table, the per-client injection time for **TraMark** is approximately 0.67 seconds, which is comparable to FedTracker (0.35 seconds). For a system with 10 clients, the total injection time is only 6.85 seconds per round. Given that cross-silo FL typically involves significant latency from local training and network communication, this server-side overhead is negligible in practice.

Table 3: Computational time (seconds) for the watermarking process of WAFFLE, FedTracker, and **TraMark**.

| Method | Time (s) |
|---|---|
| WAFFLE | 6.15 |
| FedTracker | 3.63 |
| FedTracker (per-client) | 0.35 |
| **TraMark** | 6.85 |
| **TraMark** (per-client) | 0.67 |

**Additional Discussions.** We have included further analyses to address key aspects of practical deployment. Specifically, Appendix A.9 examines the scalability of **TraMark**, where new large-scale experiments confirm that our method maintains robust performance as the system scales. In Appendix A.11, we analyze the communication overhead, demonstrating that **TraMark** does not impose additional bandwidth costs compared to existing traceable watermarking methods. Appendix A.12 presents a detailed breakdown of per-client MA, confirming that all clients exhibit uniform performance with negligible disparity. Finally, Appendix A.13 discusses the critical trade-off between traceability and client anonymity, arguing that while traceability reduces model-level anonymity, it is a necessary condition for accountability in secure FL settings where the model leaker is a serious concern.

## 6 CONCLUSION

We formalize the problem of injecting traceable black-box watermarks in FL. Based on the problem, we propose **TraMark**, which creates a personalized, traceable watermarked model for each client. **TraMark** first constructs a personalized global model for each client via masked aggregation. Subsequently, the watermarking process is exclusively performed in the watermarking region of each model using a distinct watermark dataset. The personalized watermarked models are then sent back to each client for local training or deployment. Extensive experiments demonstrate the effectiveness of **TraMark** across diverse FL settings. We further show that the parameters in the watermarking region are highly coupled with those in the main task region, making the watermarks robust against attacks. We also conduct a comprehensive hyperparameter study of **TraMark**.

## ACKNOWLEDGMENT

The work of Jiahao, Rui, and Zikai was supported by the National Science Foundation under the Harnessing the Data Revolution for Nevada Fire Science (HDRFS) Seed Grant NSHE-24-37. This material is based upon work co-supported by the U.S. Department of Energy, Office of Science, Office of Advanced Scientific Computing Research under Contract No. DE-AC05-00OR22725. This manuscript has been co-authored by UT-Battelle, LLC under Contract No. DE-AC05-00OR22725 with the U.S. Department of Energy. The United States Government retains and the publisher, by accepting the article for publication, acknowledges that the United States Government retains a non-exclusive, paid-up, irrevocable, world-wide license to publish or reproduce the published form of this manuscript, or allow others to do so, for United States Government purposes. The Department of Energy will provide public access to these results of federally sponsored research in accordance with the DOE Public Access Plan (http://energy.gov/downloads/doe-public-access-plan).

## ETHICS STATEMENT

This work strictly adheres to the ICLR Code of Ethics.

## REPRODUCIBILITY STATEMENT

We have made every effort to ensure the reproducibility of our results. The implementation of our method is available in a GitHub repository (`https://github.com/JiiahaoXU/TraMark`).

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

# A APPENDIX

## A.1 NOTATION TABLE

Table 4: Notation table.

| Symbol | Description |
|---|---|
| $n$ | The number of local clients |
| $\theta$ | The DNN model |
| $\theta_i$ | The model of client $i$ |
| $\theta^t$ | The model parameter vector at training round $t$ |
| $\theta'$ | The watermarked model |
| $\tilde{\theta}_i$ | The masked aggregated global model for client $i$ |
| $\tilde{\theta}_i^s$ | The $\tilde{\theta}_i$ at watermarking iteration $s$ |
| $g_i^s$ | The gradient of $\tilde{\theta}_i^s$ at watermarking iteration $s$ |
| $d$ | The dimension of $\theta$ |
| $F_i(\cdot)$ | The local learning objective of client $i$ |
| $\mathcal{D}_i^l$ | The local dataset of client $i$ |
| $\mathcal{L}(\cdot)$ | The loss function |
| $(z, y)$ | The datapoint sampled from a dataset |
| $\tau_l$ | The local training iterations |
| $\Delta_i^t$ | The local model update of client $i$ at training round $t$ |
| $\mathcal{D}^w$ | The whole watermarking dataset |
| $x$ | The backdoor trigger |
| $\mathcal{X}^w$ | The watermarking trigger set |
| $\phi(x)$ | The unique predefined output distribution for $x \in \mathcal{X}^w$ |
| $\delta$ | A valid black-box watermark |
| $\delta_i$ | A valid black-box watermark for client $i$ |
| $\mathbf{y}(\theta', x)$ | The output probability distribution of $\theta'$ given $x$ |
| $\texttt{Div}(\cdot)$ | The divergence measurement function |
| $\sigma$ | A predefined collision threshold |
| $k$ | The partition ratio |
| $\mathbf{M}_w$ | The watermarking mask |
| $\mathbf{M}_m$ | The main task mask |
| $\eta_w$ | The watermarking leaning rate |
| $\tau_w$ | The watermarking iteration |
| $\alpha$ | The warmup training ratio |

## A.2 RELATED WORK

Protecting the IP of FL models has been extensively studied recently, with most existing approaches leveraging either parameter-based (Uchida et al., 2017; Darvish Rouhani et al., 2019; Chen et al., 2023; Liang & Wang, 2023; Li et al., 2022b; Zhang et al., 2024; Yu et al., 2023; Xu et al., 2024) or backdoor-based watermarking (Tekgul et al., 2021; Liu et al., 2021; Li et al., 2022a; Shao et al., 2024; Luo & Chow, 2024; Nie & Lu, 2024b;a; Wu et al., 2022). While both approaches aim to verify model ownership, backdoor-based methods are more practical as they do not require access to model parameters. However, ensuring traceability, i.e., identifying the specific source of a leaked model, remains an open challenge for backdoor-based watermarks.

**Parameter-based Watermarking.** Parameter-based watermarking methods typically embed cryptographic information directly into the parameter space of the global model. For example, Uchida et al. (2017) proposed the first watermarking method for DNNs by incorporating a regularization loss term to embed a watermark into the model weights. Similarly, FedIPR (Li et al., 2022a) embeds messages in the Batch Normalization layers by assigning each client a random secret matrix and a designated embedding location. However, during verification, the verifier must access the model parameters to extract the embedded information. Consequently, these approaches assume that the verifier has full access to the suspect model, which is often unrealistic in real-world scenarios where leaked models may be only partially accessible (e.g., via API queries) (Lansari et al., 2023).

Table 5: Comparison of watermarking methods in FL. We evaluate whether each method supports black-box traceability verification (**BB**), operates entirely on the server side (**SS**), and enables traceability of the local model (**Traceable**). Our method, **TraMark**, is the only approach that satisfies all three properties.

| Method | BB? | SS? | Traceable? |
|---|---|---|---|
| WAFFLE (Tekgul et al., 2021) | ✓ | ✓ | ✗ |
| FedIPR (Li et al., 2022a) | ✓ | ✗ | ✗ |
| FedTracker (Shao et al., 2024) | ✗ | ✓ | ✓ |
| RobWe (Xu et al., 2024) | ✗ | ✓ | ✓ |
| FedCRMW (Nie & Lu, 2024a) | ✓ | ✗ | ✓ |
| **TraMark** (Ours) | ✓ | ✓ | ✓ |

**Backdoor-based Watermarking.** Backdoor-based watermarking has been explored as a more practical alternative, as it does not require direct access to the model's internal parameters. These methods leverage backdoor injection techniques to ensure that the model learns a specific trigger. A watermarked model outputs predefined responses when presented with inputs containing the trigger (Adi et al., 2018). For instance, WAFFLE (Tekgul et al., 2021) generates a global trigger dataset and fine-tunes the global model on it in each training round, thereby embedding the trigger into the model. Similarly, Liu et al. (2021) assume the presence of an honest client in the system and injects a trigger set (constructed by sampling Gaussian noise) through local training. While these methods enable black-box verification, their watermarked model lacks traceability. Moreover, some approaches rely on client-side trigger injection, which poses a high risk of exposure if a malicious client becomes aware of the process.

**Traceability of Watermarked Models.** To ensure the traceability of watermarks, Yu et al. (2023) propose replacing the linear layer of a suspect model with a verification encoder that produces distinct responses if the model originates from the FL system. FedTracker (Shao et al., 2024) extends WAFFLE by injecting a trigger into the global model while embedding local fingerprints (key matrices and bit strings) for individual clients. However, their traceability verification process requires white-box access to the suspicious model's parameters, which is generally infeasible in practice. A recent work, RobWe (Xu et al., 2024), follows a similar workflow to ours, splitting the network into two parts: one for model utility and another for embedding watermarks (key matrices). However, the model leaker can skip the watermarking step, weaken the embedded watermark, or embed a forged watermark during local training, making the client-side method less practical than the server-side method under our attack model, where all clients are potential model leakers. Another client-side method FedCRMW (Nie & Lu, 2024a), proposes a collaborative ownership verification method that indicates the leaker by the consensus of results of multiple watermark datasets. However, the watermark dataset used by FedCRMW is constructed based on the main task dataset, which incurs data privacy risks. A recent method, MFL-Owner (Gai et al., 2025), targets multi-modal FL by leveraging each client's visual and language encoders to construct an orthogonal transformation on the client's trigger set, which serves as a watermark. However, this approach suffers from poor generalizability and fails to scale effectively to broader FL systems. We further summarize the characteristics of existing methods in Table 5.

## A.3 ALGORITHM OF FEDAVG WITH **TraMark**

The full procedure of integrating FedAvg with **TraMark** is presented in Algorithm 2. During the initial warmup phase, the server performs standard FedAvg training. Upon completion of warmup, the server computes both the watermarking mask and the main task mask based on the current model parameters, using the selection ratio parameter $k$. In our implementation, we employ `torch.topk` from PyTorch to achieve the **Topk** operation in the `SelectingWMRegion` function. The training then transitions to the **TraMark** phase, as detailed in Algorithm 1.

## A.4 ALGORITHM OF MODEL LEAKER VERIFICATION

We present the algorithm for the verifier to verify a leaked model, $\theta'$, in Algorithm 3. Specifically, given a leaked model $\theta'$ associated with a pre-assigned label $i$, where client $i$ is the suspected leaker,

---

**Algorithm 2:** FedAvg with **TraMark**

---

   **Input** : number of clients $n$, local learning rate $\eta_l$, local iterations $\tau_l$, warmup rounds $t'$, total training rounds $T$, watermark dataset $\{\mathcal{D}_i^w\}_{i=1}^n$, partition ratio $k$, watermarking learning rate $\eta_w$, watermarking iteration $\tau_w$.

   **Output:** $\{\theta_i^T\}_{i=1}^n$.

**1**  **Initialization:** Initialized model $\theta^0 \in \mathbb{R}^d$

**2**  **Function** `LocalTraining`$(\theta)$:

**3**     **for** $s = 0$ **to** $\tau_l - 1$ **do**

**4**         $g_i^s \leftarrow \nabla_\theta \mathcal{L}(\theta^s; \mathcal{D}_i^l)$

**5**         $\theta_i^{s+1} \leftarrow \theta_i^s - \eta_l g_i^s$

**6**     **end**

**7**     **return** $\theta_i^{\tau_l} - \theta$

**8**  **Function** `SelectingWMRegion`$(\theta, k)$:

**9**     $\mathbf{M}_m \leftarrow \mathbf{0}^d$

**10**    topk_idx $\leftarrow \mathbf{TopK}(\theta, k)$ ;                            // torch.topk

**11**    $\mathbf{M}_m[\text{topk\_idx}] \leftarrow 1$

**12**    $\mathbf{M}_w \leftarrow \mathbf{1}^d - \mathbf{M}_m$

**13**    **return** $\mathbf{M}_w, \mathbf{M}_m$

**14** $\theta_i^0 \leftarrow \theta^0, \ \forall i \in [n]$

**15** **for** $t = 0$ **to** $T - 1$ **do**

**16**    Broadcast $\theta_i^t$ to each client $i$

**17**    **for** each $i \in [n]$ **in parallel do**

**18**       $\Delta_i^t \leftarrow$ `LocalTraining`$(\theta_i^t)$

**19**    **end**

**20**    **if** $t < t' - 1$ **then**

       // **FedAvg (warmup training)**

**21**       $\theta_i^{t+1} \leftarrow (1/n) \sum_{i=1}^n (\theta_i^t + \Delta_i^t), \ \forall i \in [n]$

**22**    **else**

**23**       **if** $t == \alpha \times T$ **then**

**24**         $\mathbf{M}_w, \mathbf{M}_m \leftarrow$ `SelectingWMRegion`$(\theta^t, k)$

**25**       **end**

      // **TraMark** process

**26**       $\{\theta_i^{t+1}\}_{i=1}^n \leftarrow \mathbf{TraMark}\left(\{\theta_i^t, \Delta_i^t\}_{i=1}^n, \{\mathcal{D}_i^w\}_{i=1}^n, \mathbf{M}_m, \mathbf{M}_w, \eta_w, \tau_w\right)$

**27**    **end**

**28** **end**

**29** **Return** $\{\theta_i^T\}_{i=1}^n$

---

---

**Algorithm 3:** Model Leaker Verification

---

   **Input** : A potentially leaked model $\theta'$ suspected to belong to client $i$, where $i$ is the assigned label; watermarking test set $\mathcal{D}_{\text{test}}^w$.

   **Output:** Verification result.

**1** acc $\leftarrow$ `calculate_per_label_accuracy`$(\theta', \mathcal{D}_{\text{test}}^w)$

**2** **if** $i = \arg\max_j \text{acc}[j]$ **then**

**3**    **return** Verification successful

**4** **else**

**5**    **return** Verification failed

**6** **end**

---

the verifier evaluates $\theta'$ on the full test set and computes the per-label accuracy (Line 1 in Algorithm 3). The verifier then selects the label with the highest accuracy. If this label matches the assigned label $i$ (Line 2), the verification is considered successful (Line 3); otherwise, it is considered a failure (Line 5). Since our attack model assumes any local client could be a potential leaker, we apply the verification process to each watermarked model. VR is then defined as the percentage of watermarked models successfully verified.

## A.5 HARDWARE SETTINGS

All experiments were carried out on a self-managed Linux-based computing cluster running Ubuntu 20.04.6 LTS. The cluster is equipped with eight NVIDIA RTX A6000 GPUs (each with 48 GB of memory) and AMD EPYC 7763 CPUs featuring 64 cores.

## A.6 OUTPUT DIVERGENCE

We provide empirical evidence demonstrating how **TraMark** effectively prevents watermark collisions. Recall that in Problem 1, the constraint is designed to maximize the divergence between the outputs of different models when given the same inputs, thereby mitigating the risk of watermark collisions. To illustrate this, we compute the KL divergence between each watermarked model and all other watermarked models on

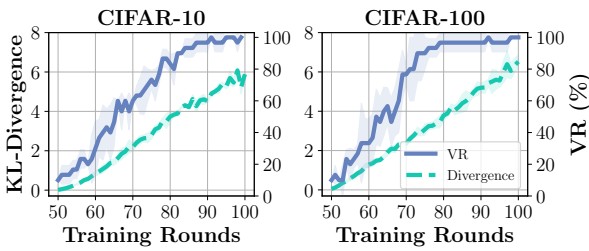

Figure 5: The averaged KL divergence and VR over training rounds on CIFAR-10 and CIFAR-100 datasets.

the respective watermark test set. We plot the average KL divergence and VR for CIFAR-10 and CIFAR-100 datasets in Figure 5. The results clearly show a consistent increase in KL divergence as training progresses. This trend arises because, as the watermarking injection process continues, each watermarked model refines its unique watermark patterns, making it more distinguishable from others. Consequently, the VR also increases, further confirming the effectiveness of **TraMark** in preventing watermark collisions.

## A.7 LARGE-SCALE FL WITH CLIENT SAMPLING

Table 6: Performance of **TraMark** under different client sampling settings in large-scale FL. MAs and VRs are shown in percentages (%).

| Method | Tiny-ImageNet | | Tiny-ImageNet (CS) | |
|---|---|---|---|---|
| | MA | VR | MA | VR |
| **TraMark** | 17.32 | 100 | 17.07 | 100 |

Here, we evaluate the effectiveness of **TraMark** in a large-scale FL setting with 50 local clients. We primarily consider two scenarios: FL without client sampling and FL with *client sampling* (CS). In the client sampling scenario, the server randomly selects 20% of the clients in each training round to perform local training. For **TraMark**, we enforce the injection of watermarks for all clients in each round, regardless of whether they are sampled or not. We use WafflePattern as the source for the watermark dataset, as it allows for generating an arbitrary number of distinct classes. Our experiments are conducted on the Tiny-ImageNet dataset, and the results are summarized in Table 6. The results show that **TraMark** consistently ensures a complete VR in both scenarios. This demonstrates the strong generalization ability of **TraMark** across different client settings.

## A.8 MORE RESULTS ON FINE-TUNING ATTACK, QUANTIZATION ATTACK, AND ADAPTIVE ATTACK

Table 7: Impact of model quantization on MA and VR, comparing FP16 and INT8 against the FP32 baseline. MAs and VRs are shown in percentages (%).

| Dataset | FP32 (Baseline) | | FP16 | | INT8 | |
|---|---|---|---|---|---|---|
| | MA | VR | MA | VR | MA | VR |
| FMNIST | 91.20 | 96.67 | 91.94 | 96.67 | 91.92 | 96.67 |
| CIFAR-10 | 88.58 | 100.00 | 88.35 | 100.00 | 88.35 | 100.00 |
| CIFAR-100 | 61.13 | 100.00 | 60.99 | 96.67 | 60.99 | 96.67 |
| Tiny-ImageNet | 20.91 | 100.00 | 20.20 | 100.00 | 20.19 | 100.00 |
| **Average** | 67.46 | 99.17 | 65.37 | 98.34 | 65.36 | 98.34 |

**Quantization Attack.** We assume that malicious clients may quantize their local models to impact the effectiveness of watermarks. Following (Shao et al., 2024), we conduct experiments on

watermarked models trained by **TraMark** that are quantized to FP16 and INT8. The MA and VR results are summarized in Table 7. Compared to the FP32 baseline, quantizing the model to FP16 and INT8 leads to a 2.09% and 2.10% drop in MA, respectively, and a 0.83% drop in VR. The negligible decrease in VR demonstrates the robustness of the watermarks injected by **TraMark** against quantization attacks.

**More Results on Fine-tuning Attack.** By default, we set the fine-tuning learning rate to 0.01 (local training learning rate), consistent with the training phase. We gradually increase the fine-tuning learning rate to an extreme value of 0.1 and perform fine-tuning for 30 rounds. The resulting MA and VR are shown in Figure 6. Our results demonstrate that **TraMark** exhibits strong resilience against fine-tuning attacks: as long as MA remains stable, VR is preserved. When the fine-tuning learning rate becomes excessively large and causes a drop in MA, VR also decreases accordingly. This indicates that the two regions are tightly coupled, making it difficult for malicious clients to remove the embedded watermarks through fine-tuning.

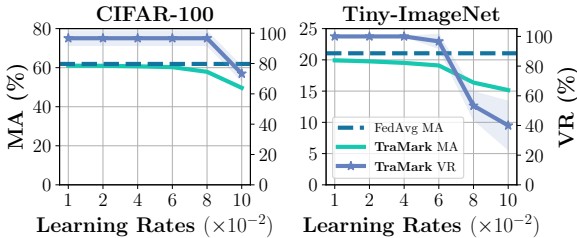

Figure 6: MA, VR changes with various fine-tuning learning rates on CIFAR-100 and Tiny-ImageNet datasets.

Table 8: Performance of **TraMark** under RNP. MAs and VRs are shown in percentages (%).

| Method | FMNIST | | CIFAR-10 | | CIFAR-100 | |
|---|---|---|---|---|---|---|
| | MA | VR | MA | VR | MA | VR |
| **TraMark** | 91.20 | 96.67 | 88.58 | 100.00 | 61.13 | 100.00 |
| RNP | 73.46 | 70.00 | 85.08 | 90.00 | 60.57 | 100.00 |

**Adaptive Attack.** We consider a more challenging scenario where malicious clients are aware that the received global model has been embedded with a black-box watermark. As a result, they attempt to remove the watermark using a backdoor removal method. We adopt Reconstructive Neuron Pruning (RNP) (Li et al., 2023), a state-of-the-art backdoor removal technique, for this purpose. Since RNP requires a Batch normalization layer while ViT employs Layer normalization, we conduct experiments on FMNIST, CIFAR-10, and CIFAR-100. The MA and VR results of RNP on the watermarked models are summarized in Table 8. We observe that RNP has limited effectiveness on the FMNIST and CIFAR-10 datasets, where the VR decreases from 90% and 100% to 70% and 90%, respectively. However, in these cases, the MA also drops significantly. For CIFAR-100, RNP has no impact on the VR but still leads to a degradation in MA. These results highlight the robustness of the watermarks embedded in each global model, demonstrating the strong effectiveness of **TraMark**.

## A.9 SCALABILITY ANALYSIS

To evaluate the practical applicability of **TraMark** across varying system scales, we discuss its scalability from three critical perspectives: watermark dataset preparation, label space capacity, and computational overhead. We further support this analysis with large-scale experiments on ImageNet-1k (Deng et al., 2009).

**Watermark Dataset Scalability.** Theoretically, **TraMark** requires $n$ distinct OOD datasets for $n$ clients to ensure unique traceability. In practice, this requirement poses no barrier to scalability. As demonstrated in our experiments, **TraMark** is fully compatible with synthetic pattern generation methods (e.g., WafflePattern). This approach allows for the automated, zero-cost generation of an effectively infinite number of unique trigger sets, thereby eliminating the need to collect or curate real-world OOD data for each new client.

**Label Space Scalability via Virtual Class Extension.** TraMark requires each client to have a dedicated output slot. Our work focuses on cross-silo FL settings such as collaborations among hospitals or financial institutions, where the number of clients is usually small (for example, < 20). In such cases, assigning a unique output position to each client is straightforward and adds no structural or operational difficulty. When the number of clients grows beyond the size of the existing output layer, the model can be expanded before FL training begins, for instance, by adding output

neurons. We conduct an experiment where the classifier head is enlarged from 10 to 20 outputs on CIFAR-10. TraMark maintains full VR with only a $0.34\%$ drop in main accuracy compared to FedAvg, showing that the method remains stable when the output space is extended. This confirms that expanding the label space is a stable operation that does not compromise model utility.

**Computational Scalability.** The server-side watermark injection process scales linearly with the number of participating clients ($\Theta(n)$). While simulating thousands of clients is computationally intensive, we emphasize that **TraMark** is primarily targeted at cross-silo FL settings. In such settings, the linear increase in server-side computation is highly efficient and remains negligible compared to the communication and local training latencies.

**Large-scale Validation on ImageNet-**$1\mathbf{k}$**.** To empirically demonstrate scalability beyond standard benchmarks, we conducted a large-scale experiment on the ImageNet-1k dataset with 100 clients. We utilized the `vit-base-patch16-224-in21k` (Dosovitskiy et al., 2021) foundation model and WafflePattern triggers. Over 60 training rounds, the system achieved an average MA of $77.80\%$ and a full $100\%$ VR. These results confirm that **TraMark** maintains robust performance and traceability even when scaling to larger client pools and complex model architectures.

## A.10 More Discussions and Future Directions

**Malicious Client Collusion.** In our work, we adopt the benchmark FL paradigm, FedAvg, where each local client receives an identical model. However, since **TraMark** injects watermarks within the designated watermarking region, malicious clients may collude to identify its location by comparing their identical main task parameters. This limitation can be solved by leveraging personalized FL (Tan et al., 2022; T Dinh et al., 2020; Wu et al., 2020), where the server assigns unique model weights to each client, ensuring distinct watermarked models and enhancing security.

**Computational Overhead with Verification.** While we assume the server possesses sufficient computational resources, practical constraints require a careful evaluation of the maximum potential overhead. Having discussed the isolated injection cost in the previous section, we report the cumulative server-side time on CIFAR-10 in Table 9. This metric encompasses model aggregation, watermark injection, and watermark verification. Under this comprehensive metric, **TraMark** exhibits a higher total execution time ($74.35$s). However, as emphasized in our defense model, watermark verification is an on-demand audit step performed only when leakage is suspected or at the end of training. Consequently, this value represents a worst-case scenario and should not be interpreted as the standard per-round latency incurred during normal training.

Table 9: Comparison of full server-side computational time (seconds). The reported value includes the time for aggregation, watermark injection, and verification. Note that FedAvg involves only aggregation.

| Method | FedAvg | WAFFLE | FedTracker | TraMark |
|---|---|---|---|---|
| Total Time (Agg. + Inject. + Verify.) | 1.03 | 12.03 | 13.32 | 74.35 |

**False Positive Cases.** A common challenge faced by black-box watermarking methods is the occurrence of false positives. Specifically, given an unwatermarked model, a black-box watermarking method may still output a prediction, which has a chance of being incorrectly interpreted as watermarked. For our proposed method, **TraMark**, the expected false positive rate is $1/r$, where $r$ is the number of output labels of the model. Consequently, for large-scale datasets such as Tiny-ImageNet, which contains 200 classes, the false positive rate is negligible. However, for smaller datasets, the issue becomes more pronounced. Addressing false positives in small-scale datasets is an important direction for future work.

**Theoretical Analysis.** Although **TraMark** demonstrates strong empirical performance, there remains a gap in providing a theoretical guarantee for ensuring the traceability of watermarked models. We leave this theoretical analysis as future work.

## A.11 Discussion on Communication Overhead

We analyze the communication overhead of **TraMark** in practical network deployments, specifically distinguishing between cross-silo and cross-device settings. In our primary target scenario, cross-silo FL (e.g., collaborations between institutions), systems typically utilize standard TCP/IP

protocols where the server communicates with each client via individual point-to-point (unicast) connections. In this context, transmitting $n$ personalized models to $n$ clients consumes the exact same bandwidth as transmitting one identical global model $n$ times, as the total data volume remains unchanged. Since **TraMark** modifies only parameter values without altering the model architecture or size, it introduces no additional communication load in cross-silo deployments.

In broadcast-capable environments (e.g., cross-device FL), while distributing distinct models theoretically reduces broadcast efficiency, we argue that this overhead is justifiable for two reasons. First, model differentiation is a fundamental prerequisite for traceability; identifying a specific leaker inherently requires unique model instances, which is a requirement of the traitor tracing problem rather than a limitation specific to **TraMark**. Second, in settings where multiple devices belong to a single user, per-device watermarking is unnecessary, allowing the same watermarked model to be shared across all devices owned by the same identity. Finally, **TraMark** maintains a communication cost profile consistent with existing state-of-the-art traceable methods like FedTracker, which similarly rely on distributing personalized model states.

## A.12 PERFORMANCE FAIRNESS ACROSS CLIENTS

In the main text, we reported the average MA to evaluate the utility of the watermarked models. Here, we provide a detailed breakdown of the per-client accuracy. Specifically, we report the individual testing accuracy of the personalized models for all participating clients (Client ID 0–9) on both CIFAR-10 and CIFAR-100 datasets. The experiments were conducted with a fixed random seed (0). The results are summarized in Table 10.

Table 10: Per-client MA on CIFAR-10 and CIFAR-100. The results demonstrate high uniformity across all participants.

| Dataset | ID: 0 | ID: 1 | ID: 2 | ID: 3 | ID: 4 | ID: 5 | ID: 6 | ID: 7 | ID: 8 | ID: 9 |
|---|---|---|---|---|---|---|---|---|---|---|
| **CIFAR-10** | 88.48 | 88.69 | 88.33 | 88.36 | 88.08 | 88.40 | 88.13 | 88.30 | 88.37 | 88.14 |
| **CIFAR-100** | 61.24 | 61.40 | 60.79 | 60.85 | 61.22 | 61.06 | 60.89 | 61.35 | 60.99 | 61.12 |

As shown in the table, the performance disparity among clients is negligible. On the CIFAR-10 dataset, the standard deviation of accuracy across clients is approximately $0.17\%$, while on the more complex CIFAR-100 dataset, it remains low at approximately $0.19\%$. We observed no outlier cases where a specific client received a poorly performing model. These results confirm that **TraMark** preserves equitable model utility for all participants in the federated system.

## A.13 DISCUSSION ON THE TRADE-OFF BETWEEN PRIVACY AND TRACEABILITY

A key motivation for using FL is preserving data privacy. Adding traceability raises questions about whether these guarantees still hold. Here, we discuss this trade-off in **TraMark** and show that although model-level anonymity is reduced, data privacy remains fully preserved.

In standard FL systems such as FedAvg, privacy comes from keeping raw data local while sharing only model updates. **TraMark** follows the same process. The server already receives separate updates from each client, and our method does not change this communication protocol or access any local data. The traceability mechanism is based only on OOD triggers generated by the server, which do not reveal information about private training data.

The trade-off introduced by **TraMark** concerns model anonymity. In settings that use secure aggregation, the server observes only the aggregated update and cannot identify individual contributions. In contrast, **TraMark** requires distributing and verifying personalized models, which means the server knows which model belongs to which client. This reduction in anonymity is a functional requirement for accountability. In many cross-silo FL applications, the server is the model owner and must be able to identify model leakage to protect intellectual property. In such cases, full anonymity would prevent accountability and allow model leakers to go untraceable.

## A.14 USE OF LLM STATEMENT

We used LLM solely for grammar checking and polishing the writing of this manuscript.

