# OpenReview forum: "Traceable Black-Box Watermarks For Federated Learning"
_ICLR.cc/2026/Conference — ICLR 2026 Poster_

### Official Review · Reviewer_LhGb · 2025-10-27

**Soundness:** 3
**Presentation:** 3
**Contribution:** 2
**Rating:** 6
**Confidence:** 3

**Summary:**

This paper presents a method to address the copyright issue in Federated Learning (FL) systems where clients might leak the shared global model. The authors introduce TraMark, a server-side watermarking framework designed to be both black-box (verifiable without parameter access) and traceable (capable of identifying the specific client who leaked the model). The core mechanism involves partitioning the model's parameter space into a 'main task region' and a 'watermarking region', identifying the latter by selecting the least important parameters after a warmup phase. Using a "masked aggregation" process, the server combines updates from all clients only in the main task region. It simultaneously preserves a distinct watermarking region for each individual client, into which a unique watermark is injected using a distinct dataset. This approach provides each client with a personalized model that maintains high performance on the primary task while embedding a unique identifier, crucially preventing the watermarks from being destroyed during the aggregation process. The authors' evaluation shows the method achieves high traceability with a minimal drop in main task accuracy and demonstrates robustness against removal attacks like pruning and fine-tuning.

**Strengths:**

- This paper has a clear motivation to achieve black-box traceability in FL.
- This paper provides a comprehensive evaluation of various datasets and both IID and non-IID settings.
- This paper is generally well-written and easy to follow.

**Weaknesses:**

- Insufficient ablation study. To better justify the necessity of the parameter partitioning, it may be better for the authors to include a comparison against a simpler baseline that does not use this partitioning. This would help quantify the impact of watermark collisions or main task degradation that the partitioning scheme is designed to prevent.
- Inadequate evaluation of computational overhead. The proposed method requires the server to perform personalized watermark injection (fine-tuning) for every client in each round. This process could be much slower than white-box methods like FedTracker, especially as the number of clients increases. It may be better for the authors to evaluate the computational time cost with a varying number of clients.
- Lack of clarity in reported results. Some reported data is imprecise. For example, in Table 1, the per-dataset Verification Rate (VR) is only represented by a checkmark (if >95%) or an 'X', with only an average VR reported. It may be better for the authors to report the specific VRs for all methods and datasets to allow for a more granular and direct comparison.

**Questions:**

Please refer to the Weaknesses.

---

> ### Author Response · Authors · 2025-11-21
> **Responses to Reviewer LhGb (1/1)**
>
> Dear reviewer LhGb,
>
> **We sincerely appreciate the reviewer's time to carefully read our work and give us comments and suggestions for improving the quality and completeness of our manuscript. Here, we kindly address your questions and concerns as follows.**
>
> ---
>
> > **[W1] Insufficient ablation study. To better justify the necessity of the parameter partitioning, it may be better for the authors to include a comparison against a simpler baseline that does not use this partitioning. This would help quantify the impact of watermark collisions or main task degradation that the partitioning scheme is designed to prevent.**
>
> **Response.** We thank the reviewer for this constructive suggestion. We agree that quantifying the impact of parameter partitioning is essential to justify our design choices.
>
> To address this, we conducted an ablation study on CIFAR-10 by comparing TraMark against a baseline, TraMark-w/o-P (TraMark without partitioning). In this baseline, the server injects the watermark into the full parameter space without isolating a dedicated region. We also report the verification leakage, defined as the average of the average accuracy of a specific client's model on other clients’ watermark triggers.
>
>
> | Dataset | MA (%) | VR (%)  | Verification Leakage (%) |
> |:-:|:-:|:-:|:-:|
> | TraMark-w/o-P | 87.28 | 80.00 | 39.75 |
> | TraMark | 88.58  | 100.00 | 4.35 |
>
> TraMark-w/o-P yields a lower MA (87.28%) compared to TraMark (88.58%). This confirms that without partitioning, the watermark injection directly interferes with the main task weights, leading to performance degradation. Crucially, TraMark-w/o-P exhibits significant verification leakage (39.75%), meaning that models frequently misfire on triggers intended for other clients. In contrast, TraMark reduces this leakage to 4.35%, ensuring that watermarks remain unique and identifiable.
>
> These results quantitatively justify the necessity of parameter partitioning. It effectively decouples the watermarking objective from the main task, preventing both performance degradation and watermark collisions.
>
> ---
>
> > **[W2] Inadequate evaluation of computational overhead. The proposed method requires the server to perform personalized watermark injection (fine-tuning) for every client in each round. This process could be much slower than white-box methods like FedTracker, especially as the number of clients increases. It may be better for the authors to evaluate the computational time cost with a varying number of clients.**
>
> **Response.** We thank the reviewer for raising the concern regarding computational overhead. We have evaluated the pure watermark injection time. The results are summarized in the table below (unit: seconds):
>
> | WAFFLE | FedTracker | FedTracker (per-client) | TraMark | TraMark (per-client)|
> |-|-|-|-|-|
> |6.15| 3.63 | 0.35 | 6.85 | 0.67|
>
> As shown, the per-client injection time for TraMark is only 0.67s, which is highly efficient and comparable to the SOTA white-box method FedTracker (0.35s). Both TraMark and FedTracker's computational cost scales linearly with the number of clients. We have revised our manuscript by adding Table 3 to explicitly break down these costs.
>
> ---
>
> > **[W3] Lack of clarity in reported results. Some reported data is imprecise. For example, in Table 1, the per-dataset Verification Rate (VR) is only represented by a checkmark (if >95\%) or an 'X', with only an average VR reported. It may be better for the authors to report the specific VRs for all methods and datasets to allow for a more granular and direct comparison.**
>
> **Response.** We thank the reviewer for this helpful suggestion regarding the clarity of our results. We agree that reporting specific numerical values is essential for a precise and granular evaluation.
>
> Accordingly, we have revised Table 1 in the manuscript to explicitly list the exact VR percentages for all methods across each dataset.
>
> ---
>
> **We sincerely thank the reviewer again for your time and constructive comments. We hope these clarifications and additional results have fully resolved your concerns, and we remain open to any further discussion.**
>
> ---

---

### Official Review · Reviewer_UwiB · 2025-10-30

**Soundness:** 2
**Presentation:** 3
**Contribution:** 2
**Rating:** 4
**Confidence:** 4

**Summary:**

This paper addresses the challenge of verifying model ownership and tracing leakage in Federated Learning under a black-box setting. The authors formalize the problem and propose a framework named TraMark, which leverages parameter space partitioning and masked aggregation. Experimental results demonstrate the framework's verification rate and its impact on main task performance, complemented by a hyperparameter analysis. Furthermore, the method exhibits robustness against attacks such as pruning and fine-tuning.

**Strengths:**

A clear formalization of the traceable black-box watermarking problem is provided in Sec. 3.

The experimental results demonstrate an excellent Verification Rate (VR) of approximately 99.17% with a limited drop in Main-task Accuracy (MA), especially when compared to the FedTracker method.

The evaluation is conducted across multiple datasets, and the analyses of robustness, hyperparameters, and other factors are detailed.

**Weaknesses:**

As indicated in Table 8 of the appendix, the per-round computational overhead for TraMark's aggregation is over 70 times that of FedAvg. The authors rationalize this by citing the small number of clients in cross-silo scenarios, a justification that is not entirely convincing and severely limits the method's scope of application.

The paper appears to overlook the method's communication overhead. At the beginning of each training round, the server is required to send a unique, personalized model to every client. This results in a sharp increase in communication costs compared to broadcasting a single global model, which in turn restricts the method's applicability.

The method's reliance on constructing an independent, out-of-distribution watermark dataset and a unique output label space for each client is a strong assumption. In realistic Federated Learning scenarios, this requirement may not be feasible, thus limiting the method's practical scalability and applicability.

The paper uses the average accuracy across all personalized client models as the metric for MA. However, this average may conceal significant performance disparities, where some clients receive highly accurate models while others are left with poorly performing ones. This masks potential issues of unfairness in performance distribution.

The method, in its pursuit of model traceability, raises questions about its potential impact on privacy preservation—one of the core advantages of Federated Learning. A detailed discussion on this critical trade-off appears to be missing from the paper.

The paper does not discuss the method's feasibility and effectiveness in Federated Learning scenarios with asynchronous client participation. It is unclear how the proposed mechanism would perform under such conditions.

**Questions:**

See Weaknesses.

---

> ### Author Response · Authors · 2025-11-21
> **Responses to Reviewer UwiB (1/4)**
>
> Dear reviewer UwiB,
>
> **We sincerely appreciate your time in carefully reading our work and providing comments and suggestions to improve the quality of the manuscript. Below, we address your questions and concerns.**
>
> ---
>
> > **[W1] As indicated in Table 8 of the appendix, the per-round computational overhead for TraMark's aggregation is over 70 times that of FedAvg. The authors rationalize this by citing the small number of clients in cross-silo scenarios, a justification that is not entirely convincing and severely limits the method's scope of application.**
>
> **Response.** We thank the reviewer for carefully examining the computational overhead. We agree that the originally reported “70x” overhead in Table 8 seems concerning, and we would like to clarify that this number came from an overall metric that combined server aggregation, watermark injection, and watermark verification. In practice, watermark verification only happens after watermarking injection. We have revised the description of Table 8 (Table 9 in the new manuscript) and the related discussion. We also corrected the reported time for FedTracker, since the earlier value did not include its watermark verification time.
>
> 1. Clarification on Table 8: The time in the original Table 8 included three parts: *model aggregation time*, *watermark injection time*, and *watermark verification time*. In real FL deployments, verification is an audit procedure that happens only when leakage is suspected or at the end of training, not during every round. Including verification in the per-round cost led to an overestimation of the actual operational overhead.
>
> 2. Pure watermarking injection cost (Table 3 in the new manuscript): To reflect the true per-round watermarking injection cost, we removed the per-round verification. The updated results are shown below:
>
> | WAFFLE | FedTracker | FedTracker (per-client) | TraMark | TraMark (per-client)
> |-|-|-|-|-|
> |6.15| 3.63 | 0.35 | 6.85 | 0.67|
>
> The per-client watermark injection time for TraMark is 0.67s, which is close to FedTracker (0.35s). Although TraMark incurs a slightly higher cost, it is on the scale of seconds. The cost grows linearly with the number of clients because each client receives a unique watermark, but the absolute overhead remains low. For example, with 10 clients, the server adds about 6.7 seconds per round, which is minor compared with the network delay and the local training time common in cross-silo FL.
>
> ---

---

> ### Author Response · Authors · 2025-11-21
> **Responses to Reviewer UwiB (2/4)**
>
> > **[W2] The paper appears to overlook the method's communication overhead. At the beginning of each training round, the server is required to send a unique, personalized model to every client. This results in a sharp increase in communication costs compared to broadcasting a single global model, which in turn restricts the method's applicability.**
>
> **Response.** We appreciate the reviewer’s concern regarding communication overhead. We would like to clarify that in parctical FL framework, TraMark does not incur additional communication costs compared to the baseline.
>
> 1. Cross-silo FL with unicast communication: In practical FL systems across silos (typically implemented over TCP/IP via HTTP), the server communicates with clients (banks or hospitals) via individual point-to-point (unicast) connections. We have the following analysis:
>
>     1.1. Standard FL: Sending 1 identical global model to $n$ clients requires transmitting the model data $n$ times.
>
>     1.2. TraMark: Sending $n$ personalized models to $n$ clients also requires transmitting the model data $n$ times.
>
>     1.3. Because TraMark only modifies the parameter values without changing the model size, the total bandwidth usage is identical. TraMark therefore does not add communication load beyond that of standard FL algorithm.
>
> 2. Cross-device FL with broadcast communication: In a broadcast-capable environment (e.g., FL deployed across edge devices within a local area network rather than across the public Internet), distributing personalized models could reduce the efficiency of a single broadcast operation. However, this setting differs fundamentally from cross-silo FL in two ways.
>
>     2.1. First, when all devices belong to the same user (e.g., phones, tablets, laptops in a home network), model watermarking on a per-device basis is unnecessary; the models do not need to be uniquely traced.
>
>     2.2. Second, local broadcast in a LAN environment is extremely lightweight. The server can transmit multiple models with negligible overhead due to high bandwidth and low latency. Even if $n$ personalized models must be delivered to $n$ devices, the communication cost remains small in practice.
>
> 3. Cost is aligned with existing traceable watermarking methods: Current SOTA approaches such as FedTracker also rely on generating personalized local models and maintaining per-client watermark states. TraMark therefore maintains the same communication profile as the SOTA methods.
>
> We thank the reviewer for raising this point, which helped us improve the completeness of the paper, and we have added Appendix A.11 to discuss this issue in detail.
>
> ---

---

> ### Author Response · Authors · 2025-11-21
> **Responses to Reviewer UwiB (3/4)**
>
> > **[W3] The method's reliance on constructing an independent, out-of-distribution watermark dataset and a unique output label space for each client is a strong assumption. In realistic Federated Learning scenarios, this requirement may not be feasible, thus limiting the method's practical scalability and applicability.**
>
> **Response.** We thank the reviewer for raising concerns about the feasibility of constructing out-of-distribution watermark datasets and unique label spaces. We clarify that both requirements are practical and scalable in our **target settings** for the following reasons:
>
> 1. Scalability of the watermark dataset: While TraMark theoretically requires $n$ distinct OOD datasets, this is easily achieved in practice. As demonstrated in our paper, TraMark is fully compatible with synthetic pattern generation (e.g., WAFFLE). This allows for the automated, zero-cost generation of effectively infinite unique trigger sets, eliminating the need for collecting real-world OOD data.
>
> 2. Scalability of the label space: TraMark requires each client to have a dedicated output slot. Our work focuses on cross-silo FL settings such as collaborations among hospitals or financial institutions, where the number of clients is usually small (for example, <20). In such cases, assigning a unique output position to each client is straightforward and adds no structural or operational difficulty.
>
> 3. Extendability to larger client populations: When the number of clients grows beyond the size of the existing output layer, the model can be expanded before FL training begins, for instance, by adding output neurons. We conduct an experiment where the classifier head is enlarged from 10 to 20 outputs on CIFAR-10. TraMark maintains full VR with only a 0.34% drop in main accuracy compared to FedAvg, showing that the method remains stable when the output space is extended.
>
> 4. Compatibility with common model adaptation practice. Modifying the classification head is a standard practice in modern deep learning, particularly when adapting pretrained foundation models (e.g., vit-base-patch16) to new tasks. Therefore, our requirement to adjust the output layer aligns seamlessly with existing model adaptation pipelines, making it a natural and practical operation.
>
> Together, these points demonstrate that utilizing synthetic triggers and extending the label space are both scalable and operationally consistent with standard FL workflows.
>
> We thank the reviewer for the careful observation, which helped us improve the completeness of the paper, and we have updated Appendix A.9 accordingly.
>
> ---
>
> > **[W4] The paper uses the average accuracy across all personalized client models as the metric for MA. However, this average may conceal significant performance disparities, where some clients receive highly accurate models while others are left with poorly performing ones. This masks potential issues of unfairness in performance distribution.**
>
> **Response.** We thank the reviewer for raising this important concern regarding performance fairness across clients. We agree that examining the distribution of accuracy is an essential step to ensure that no client is disadvantaged.
>
> In TraMark, performance fairness is naturally preserved because the main task region covers almost the entire parameter space. These parameters are aggregated and synchronized across all clients in the same way as standard FL. In contrast, the watermarking region is extremely small and is selected from parameters that are unimportant to the main task. As a result, the small amount of client-specific parameters introduced by watermarking has little influence on the performance of each personalized model.
>
> To verify this, we report the individual accuracy for all clients (Client 0–9) on CIFAR-10 and CIFAR-100 (seed 0) after watermark injection on the server. The results are shown below:
>
> Table R1: Per-Client Accuracy on CIFAR-10 (%)
> | Client ID | 0 | 1 | 2 | 3 | 4 | 5 | 6 | 7 | 8 | 9 |
> | :---: | :---: | :---: | :---: | :---: | :---: | :---: | :---: | :---: | :---: | :---: |
> | MA | 88.48 | 88.69 | 88.33 | 88.36 | 88.08 | 88.40 | 88.13 | 88.30 | 88.37 | 88.14 |
>
> Table R2: Per-Client Accuracy on CIFAR-100 (%)
> | Client ID | 0 | 1 | 2 | 3 | 4 | 5 | 6 | 7 | 8 | 9 |
> | :---: | :---: | :---: | :---: | :---: | :---: | :---: | :---: | :---: | :---: | :---: |
>  | MA | 61.24 | 61.40 | 60.79 | 60.85 | 61.22 | 61.06 | 60.89 | 61.35 | 60.99 | 61.12 |
>
> As shown in the tables, the performance disparity is negligible. On CIFAR-10, the standard deviation of client accuracy is only $\pm0.17\%$. On CIFAR-100, the standard deviation is only $\pm0.20\%$.
>
> This analysis confirms that TraMark maintains fairness in performance across clients, and we have included the detailed breakdowns in Appendix A.12 of the revised manuscript. We thank the reviewer again for the careful observation, which helped us improve the clarity and completeness of our paper.
>
> ---

---

> ### Author Response · Authors · 2025-11-21
> **Responses to Reviewer UwiB (4/4)**
>
> > **[W5] The method, in its pursuit of model traceability, raises questions about its potential impact on privacy preservation—one of the core advantages of Federated Learning. A detailed discussion on this critical trade-off appears to be missing from the paper.**
>
> **Response.** We appreciate the reviewer for raising this important point regarding privacy preservation. We respectfully clarify that TraMark does not compromise the fundamental data privacy advantage of FL.
>
> - Data privacy is preserved: In standard FL systems like FedAvg, the core privacy benefit stems from keeping raw training data local while exchanging only model updates. TraMark strictly adheres to this principle. The server already receives separate local model updates from each client in every round. Our method does not alter this communication structure, nor does it access local data. The traceability mechanism relies solely on OOD triggers injected by the server, ensuring that no additional information about the client’s private data is exposed.
>
> - Model anonymity vs. accountability: We openly acknowledge that TraMark trades off model anonymity for traceability. In a standard FL system without any other security mechanisms like Secure Aggregation, the server implicitly knows the source of each update. TraMark leverages this visibility to explicitly embed watermarks. While this makes the identity of the model owner verifiable (to trace model leakers), it does not violate the data privacy guarantees of FL. In our target cross-silo scenarios, accountability is often a strict requirement that necessitates this level of transparency toward the central server.
>
> We thank the reviewer again for highlighting this critical aspect. We have added Appendix A.13 to include a detailed discussion on this privacy-traceability trade-off to ensure clarity for future readers.
>
> ---
>
> > **[W6] The paper does not discuss the method's feasibility and effectiveness in Federated Learning scenarios with asynchronous client participation. It is unclear how the proposed mechanism would perform under such conditions.**
>
> **Response.** We thank the reviewer for pointing out the importance of asynchronous FL scenarios. We would like to clarify that, despite being originally designed for synchronous FL systems, TraMark is inherently compatible with asynchronous client participation. This is primarily because our watermark injection mechanism operates independently of the aggregation process.
>
> To empirically verify this, we integrated TraMark with the classic FedAsync algorithm [1]. When the server receives updates from any client, it updates the global main task model immediately. Crucially, the server maintains the latest personalized watermarking parameters for each client. Regardless of a client's staleness, when that client requests a new model, the server combines the current global main task parameters with that client's specific watermarking region. We conducted experiments on CIFAR-10 using a staleness parameter of 0 (constant weight). The results are summarized below:
>
> | Method | MA (%) | VR (%) |
> |-|-|-|
> | FedAsync  | 85.93 | - |
> | TraMark | 81.82  | 100.00  |
>
> The results demonstrate that TraMark achieves a 100% VR, ensuring that all client models remain fully traceable even under asynchronous conditions. We acknowledge a slight drop in MA compared to the baseline, which we attribute to the challenge of optimizing watermarks alongside noisy asynchronous updates. However, this confirms the feasibility of the method. Optimizing the trade-off between asynchronous convergence and watermark fidelity remains a promising direction for future work.
>
> [1] Xie et al., "Asynchronous Federated Optimization," OPT 2020.
>
> ---
>
> **We sincerely thank the reviewer again for your time and constructive comments. We hope these clarifications and additional results have fully resolved your concerns, and we remain open to any further discussion.**
>
> ---

---

### Official Review · Reviewer_1fMj · 2025-10-30

**Soundness:** 2
**Presentation:** 3
**Contribution:** 3
**Rating:** 4
**Confidence:** 5

**Summary:**

In this paper, the authors propose TraMark, a black-box model watermarking technique suitable for federated learning setups where clients might be potential model leakers. TraMark works when the central server is benign and implements the watermarking procedure. TraMark verification also enables traitor tracing. TraMark restricts watermarking to a small subset of the model parameters using binary masks and adapts the weight update procedure to produce watermarked global models per each client while preserving the existence of the watermark. I think the idea is nice and simple, but the recovery of watermarked weights and possible evasion methods are not included in the paper.

**Strengths:**

S1. TraMark partitions the model parameters to the main task and watermarking task regions. In this way, we can say that TraMark is model-agnostic.

S2. The problem formulation and the insights in Section 2 are very clear. It is difficult to inject watermarks that satisfy black-box traceability while avoiding collusion, and I think the authors did a good job of formulating this difficulty.

**Weaknesses:**

W1. Potential watermark detectability by malicious clients:

Watermarked weights may be relatively easy to detect by malicious clients. They could analyze which parameters change more significantly or differently during training and identify trends in weight updates of received global models. Such differences, especially if certain parameters are updated disproportionately or the updates remain closer to zero, could reveal which weights are used to embed covert information (i.e., the watermark in this case). This issue may become even more pronounced with the warm-up phase, where clients suddenly see a change in weight updates after the warm-up. The authors should include experiments or analyses investigating this potential vulnerability.

W2. Key discussions moved to the appendix:

Several important discussion points are placed in the Appendix, without even a summary of the results in the main text. This weakens the perceived impact of the contributions. I recommend moving the experimental setup to the Appendix instead and bringing key takeaways or insights from the additional experiments currently in the appendix into the main text.

W3. Other weaknesses:

Please refer to the detailed questions below for additional points of concern.

**Questions:**

Q1. The authors should verify the correctness of the compared methods presented in the appendix. For example, FedTracker can perform ownership verification in a black-box manner, as the related paper explicitly states that "the zero-bit backdoor-based watermark is feasible for ownership verification and can be verified through black-box access." Additionally, I do not agree with the claim that RobWe is less practical because watermarking is performed on the client side. In fact, this seems more practical: each client can have their own secret watermark, eliminating potential watermark collusion problems, and maintaining robustness even when the server is untrusted.

Q2. There is insufficient discussion of scalability. The paper only tests with 10–50 clients. How would the proposed method perform with 1000 or 10000 clients? Moreover, how does the approach scale in relation to collusion resistance and the need to maintain personalized global models for each client?

Q3. The authors consider federated averaging as the only aggregation method. How would TraMark perform with alternative aggregation strategies such as Krum or other robust aggregation methods?

Q4. The format of in-text references is wrong. The authors should also be included in parentheses.

Q5. What is the model accuracy after the warm-up phase? Could this intermediate model (which includes no watermarks) be distributed directly instead of the last trained model?

Q6. The assumption that the server has full access to all local models is too strong and contradicts one of the main motivations for federated learning: privacy preservation. This assumption should be relaxed or at least discussed in detail.

Q7. Table 1 only reports the average results. The authors should also include verification rates (VR) for each method on each dataset. The current figure does not clearly convey these proportions.

Q8. Have the authors considered the possibility of out-of-distribution (OOD) detection as an evasion strategy against the proposed watermarking method? Especially considering the fact that they are using OOD samples as watermarks.

Q9. The repository link provided in the OpenReview abstract results in an error ("not found"). However, the link in the PDF appears to work.

**Details Of Ethics Concerns:**

No ethical concerns

---

> ### Author Response · Authors · 2025-11-21
> **Responses to Reviewer 1fMj (1/5)**
>
> Dear reviewer 1fMj,
>
> **We sincerely appreciate your time in carefully reading our work and providing comments and suggestions to improve the quality of the manuscript. Below, we address your questions and concerns.**
>
> ---
>
> > **[W1] Potential watermark detectability by malicious clients: Watermarked weights may be relatively easy to detect by malicious clients. They could analyze which parameters change more significantly or differently during training and identify trends in the weight updates of received global models. Such differences, especially if certain parameters are updated disproportionately or the updates remain closer to zero, could reveal which weights are used to embed covert information (i.e., the watermark in this case). This issue may become even more pronounced with the warm-up phase, where clients suddenly see a change in weight updates after the warm-up. The authors should include experiments or analyses investigating this potential vulnerability.**
>
> **Response.** We thank the reviewer for raising this insightful point regarding the potential detectability of watermarked weights by analyzing updated parameters. We address your concern as follows.
>
> In TraMark, the watermarking region is extremely small (e.g., only 1% of total parameters). The magnitude changes in these parameters is comparable to that of the main task parameters. Given the high dimensionality of modern models (millions of parameters), isolating such a sparse and statistically conformant signal is very challenging.
>
> We take this concern seriously and have conducted a new experiment to quantitatively demonstrate that detecting the watermarking region is statistically infeasible for model leakers.
>
> [**Empirical Validation.**] We analyzed the distribution of parameter updates before and after local training. Specificailly, we calculated the absolute parameter difference for each client's training ($|\Delta W| = |W_{trained} - W_{initial}|$). We then identified the parameters with the Top 1\% (largest updates) and Bottom 1\% (smallest updates) magnitudes. We measured the proportion of watermarked parameters within these two extreme groups, denoted as Top1W and Bottom1W. These metrics are calculated and averaged across all training rounds, including the warm-up training phrase.
>
> As shown in the following table, on both CIFAR-10 and CIFAR-100, watermarked parameters generated by our method account for approximately 1\% of both the Top 1\% and Bottom 1\% groups. These results align with the random baseline (since the watermark region occupies 1\% of the total space).
>
> | Dataset | Top1W (%) | Bottom1W (%) |
> |-|-|-|
> | CIFAR-10  | 1.00 | 1.00 |
> | CIFAR-100 | 1.01 | 1.00|
>
> This proves that watermarked weights do not exhibit disproportionately large or small updates compared to the rest of the model, making them statistically indistinguishable from main task weights and effectively mitigating the risk of detection.
>
> ---
>
> > **[W2] Several important discussion points are placed in the Appendix, without even a summary of the results in the main text. This weakens the perceived impact of the contributions. I recommend moving the experimental setup to the Appendix instead and bringing key takeaways or insights from the additional experiments currently in the Appendix into the main text.**
>
> **Response.** We sincerely thank the reviewer for this constructive suggestion, which has greatly helped us improve the organization and presentation quality of our paper. We fully agree that highlighting key insights in the main text is essential for demonstrating the impact of our contributions.
>
> Leveraging the additional space allowed during the rebuttal phase (up to 10 pages), we have significantly restructured the manuscript as recommended:
>
> - We have moved the analysis regarding TraMark's performance with different types of watermark datasets to the main text.
>
> - We have integrated the detailed comparison of computational overhead between TraMark and baseline methods in the main text.
>
> - We have also improved readability by providing a brief summary of the findings in the main text whenever we refer to experimental results given in an Appendix section.
>
> ---

---

> ### Author Response · Authors · 2025-11-21
> **Responses to Reviewer 1fMj (2/5)**
>
> > **[Q1] The authors should verify the correctness of the compared methods presented in the appendix. For example, FedTracker can perform ownership verification in a black-box manner, as the related paper explicitly states that "the zero-bit backdoor-based watermark is feasible for ownership verification and can be verified through black-box access." Additionally, I do not agree with the claim that RobWe is less practical because watermarking is performed on the client side. In fact, this seems more practical: each client can have their own secret watermark, eliminating potential watermark collusion problems, and maintaining robustness even when the server is untrusted.**
>
> **Response.** We thank the reviewer for their careful reading and valuable points regarding the comparison baselines. Below, we provide necessary clarifications regarding the operational settings and threat models of these methods.
>
> [**Clarification on FedTracker (white-box vs. black-box).**] We acknowledge the reviewer's correction: FedTracker indeed supports black-box ownership verification (proving the model belongs to FL).
>
> - FedTracker: While ownership verification is black-box, its traceability verification mechanism relies on embedding a unique continuous bit-string into the model parameters. Verifying this specific fingerprint to identify the leaker requires white-box access to the suspicious model’s weights.
>
> - Different from FedTracker, TraMark achieves traceability (model leaker identification) entirely in a black-box manner, requiring only API access to the suspicious model.
>
> [**Clarification on RobWe (threat model differences).**] We appreciate the reviewer's insightful perspective. We fully agree that client-side watermarking (like RobWe) is more practical and robust in scenarios where **the server is untrusted**.
>
> - Different from RobWe, our paper addresses a specific threat model where the server is the defender seeking to identify the model leakers in an FL system.
>
> - In this specific context, relying on clients to watermark themselves is fundamentally unreliable. A model leaker intending to leak the model would naturally have the incentive to skip the watermarking step, weaken the embedded watermark, or embed a forged watermark.
>
> - Therefore, for the purpose of reliable leakage attribution (where the client is the potential adversary), server-side injection is strictly necessary to guarantee that the watermark is correctly embedded into the model.
>
> We have revised the Related Work section to provide a more precise description of FedTracker and to explicitly acknowledge the value of client-side watermarking methods like RobWe in untrusted-server scenarios.
>
>
> ---

---

> ### Author Response · Authors · 2025-11-21
> **Responses to Reviewer 1fMj (3/5)**
>
> > **[Q2] There is insufficient discussion of scalability. The paper only tests with 10–50 clients. How would the proposed method perform with 1000 or 10000 clients? Moreover, how does the approach scale in relation to collusion resistance and the need to maintain personalized global models for each client?**
>
> **Response.** We sincerely thank the reviewer for raising this question regarding the scalability of our approach. We agree that a discussion on scalability is crucial for evaluating the practical applicability of our method.
>
> [**Scalability analysis.**] We discuss and address the scalability in terms of dataset preparation, output dimension, and computational cost, supported by new large-scale experiments.
>
>  1. Watermark dataset scalability: While TraMark theoretically requires $n$ distinct OOD datasets for $n$ clients, this is easily achieved in practice. As demonstrated in our paper, TraMark is fully compatible with synthetic pattern generation (e.g., WAFFLE). This allows for the automated, zero-cost generation of an infinite number of unique trigger sets, eliminating the need for collecting real-world OOD data.
>  2. Label scalability: TraMark requires a unique output slot for each client. As detailed in Appendix A.9, by expanding the final output layer (e.g., adding $N$ neurons for $N$ clients) of the model before the training process of the FL system starts, we can support a large number of clients.
>  3. Computational scalability: The watermark injection process on the server scales **linearly** with the number of clients. While simulating 1000+ clients is computationally prohibitive on our owned hardware, we emphasize that our primary target is the cross-silo FL setting (e.g., collaboration of banks or hospitals), where client counts are typically moderate (<20).
>  4. New experiment (ImageNet-1k with 100 clients): To demonstrate scalability beyond our initial settings, we conducted a new experiment on the ImageNet-1k dataset with 100 clients using the `vit-base-patch16-224-in21k` model and WafflePattern triggers. Over 60 rounds, the models achieve an average MA of 77.80% and a full VR. This shows the stable performance of TraMark when scaling to larger client pools and complex architectures.
>
>  *We remain open to the reviewer’s feedback should additional large-scale experiments be deemed necessary.* **We have added scalability analyses to Appendix A.9.**
>
> [**Collusion resistance and model maintenance.**] Recall that in Appendix A.10, we discussed countermeasures based on personalized FL models to defend against collusion attacks. We maintain that this solution is practical and effective under our default cross-silo FL settings, where the number of clients is limited. In the same time, we acknowledge that maintaining a distinct model for every client raises memory concerns as the system scales. However, by leveraging our parameter partitioning design, we can efficiently mitigate this. Since the main task region is identical across all clients, the server needs to store only a single copy of these parameters. We then store only the unique parameters of the watermarking region for each specific client. This strategy results in massive storage savings compared to full model duplication.
>
> To address the model leaker collusion for large-scale scenarios, we propose a more feasible perturbation-based defense mechanism inspired by differential privacy:
>
> - We inject small, controlled perturbations (Gaussian noise) into the main-task region of the aggregated model before distribution. The L2-norm of the perturbation is scaled to approximately 2% of the parameter vector norm.
> - This ensures that every client receives a mathematically unique "main body," preventing colluding clients from isolating the watermark region via simple weight subtraction.
> - We tested this defense on CIFAR-10 and CIFAR-100. As shown in the table below, the impact on utility is negligible, while the watermark remains fully verifiable.
>
> | Dataset | MA (%) | VR (%) |
> |-|-|-|
> | CIFAR-10  | 88.34 (0.24 $\downarrow$) | 100.00 |
> | CIFAR-100 | 61.01 (0.12 $\downarrow$) | 100.00 |
>
>
> These results confirm that TraMark can be made robust against collusion with minimal impact on performance, even in large-scale settings.
>
> ---

---

> ### Author Response · Authors · 2025-11-21
> **Responses to Reviewer 1fMj (4/5)**
>
> > **[Q3] The authors consider federated averaging as the only aggregation method. How would TraMark perform with alternative aggregation strategies such as Krum or other robust aggregation methods?**
>
> **Response.** We thank the reviewer for raising this important question regarding compatibility with robust aggregation methods. We confirm that TraMark is fully compatible with alternative aggregation strategies (such as Krum and Multi-Krum) because the watermarking mechanism is structurally decoupled from the aggregation logic of the main task.
>
> Taking Multi-Krum as a concrete example since it generally outperforms Krum, the workflow operates as follows:
>
> 1. Robust filtering: Upon receiving updates from clients, the server applies the Multi-Krum algorithm to the parameters in the min task region to identify and discard potential malicious or outlier updates.
>
> 2. Aggregation: The server aggregates only the selected "benign" updates to generate the new global parameters for the main task region.
>
> 3. Watermark injection: TraMark injects the client-specific watermarks into the watermarking region of the aggregated model before distributing it back to the clients.
>
> To validate this, we conducted experiments combining TraMark with Multi-Krum. The results are summarized below:
>
> | Dataset | MA (%) | VR (%) |
> |-|-|-|
> | CIFAR-10  | 87.85 (0.73 $\downarrow$) | 100.00 |
> | CIFAR-100 | 60.05 (1.08 $\downarrow$) | 100.00 |
>
> The results demonstrate that TraMark maintains a 100% VR, confirming that the robust aggregation process does not interfere with watermark verification. The slight decrease in MA is the side-effect of Krum-based methods (which tends to discard some benign but high-variance updates to ensure robustness).
>
> ---
>
> > **[Q4] The format of in-text references is wrong. The authors should also be included in parentheses.**
>
> **Response.** We thank the reviewer for bringing this to our attention. We have carefully revised our manuscript to confirm that we have correctly utilized the `citep` command to generate the proper citation formats.
>
> ---
>
> > **[Q5] What is the model accuracy after the warm-up phase? Could this intermediate model (which includes no watermarks) be distributed directly instead of the last trained model?**
>
> **Response.** We thank the reviewer for this question concerning the possible leakage risk of the intermediate model after the warm-up phase. For convenience, we refer to such model leakers as early-leakers. We agree that the model distributed right after warm-up does not contain watermarks.
>
> In our default setting, we assume that a model leaker would leak the model only after the full training process, which is why the warm-up ratios in the main paper are selected to maximize MA and VR. If a leaker chooses to use an intermediate model after warm-up, the leaked model is generally not useful because the accuracy is still far from convergence. For example, on CIFAR-100 with a warm-up ratio $\alpha = 0.3$, the model reaches only 43.67% MA after warm-up, while the final MA after convergence is 60.46%.
>
> To avoid this situation, the server can simply adjust the warm-up ratio $\alpha$. In settings where early-stage leakage is a concern, the server can set $\alpha$ to a smaller value (extreme case is: $\alpha = 0$). This allows the server to embed watermarks from the beginning of training, leaving no chance for an early-leaker to obtain a strong unwatermarked model. Our CIFAR-100 results support this option: with $\alpha = 0$, the system reaches a VR of 80% by round 30, with a corresponding MA of 42.75%. Although this may cause a small drop in MA, it provides higher security when early-leakers are present.
>
> ---
>
> > **[Q6] The assumption that the server has full access to all local models is too strong and contradicts one of the main motivations for federated learning: privacy preservation. This assumption should be relaxed or at least discussed in detail.**
>
> **Response.** We thank the reviewer for this valuable comment regarding the server's access to local models. We respectfully clarify that our assumption is consistent with the standard FL setup and does not contradict its original privacy benefit.
>
> 1. In the canonical FL setting (e.g., FedAvg ) where additional security mechanisms like Secure Aggregation are not explicitly deployed, the central server directly receives and aggregates local model updates. Our method operates within this standard framework where the server acts as a trusted central coordinator.
>
> 2. The primary privacy advantage of FL stems from keeping sensitive data locally: raw training data never leaves the client's device. Our method strictly adheres to this principle, as the server relies solely on model updates and never accesses private local data.
>
> We have refined our description of our defense model to state the standard FL setup we considered clearly.
>
> ---

---

> ### Author Response · Authors · 2025-11-21
> **Responses to Reviewer 1fMj (5/5)**
>
> > **[Q7] Table 1 only reports the average results. The authors should also include verification rates (VR) for each method on each dataset. The current figure does not clearly convey these proportions.**
>
> **Response.** We thank the reviewer for this helpful suggestion. Accordingly, we have made the following revisions:
>
> 1. Updated Table 1: We have refined Table 1 to explicitly list the numerical VR values for each method across all datasets, providing a precise quantitative comparison.
>
> 2. Revised Figure 1: We have updated Figure 1 by adding VRs. The clear verification interval demonstrated in the figure confirms reliable model leakage verification, supported by the consistently high VR achieved by TraMark.
>
> ---
>
> > **[Q8] Have the authors considered the possibility of out-of-distribution (OOD) detection as an evasion strategy against the proposed watermarking method? Especially considering the fact that they are using OOD samples as watermarks.**
>
> **Response.** We thank the reviewer for highlighting this potential evasion strategy. We address this concern from the perspective of our threat model and the technical characteristics of the watermark.
>
> 1. Theoretically, clients in the FL system with TraMark as the watermarking method are not aware of the watermarking process since TraMark operates completely on the server and does not require the client to do anything except local training. Consequently, they lack the prior knowledge and motivation to launch targeted detection or removal attacks.
>
> 2. Even under a stronger attack model where the adversary is aware of the OOD-based watermarking process, evasion via OOD detection is very challenging for two reasons:
>
>     2.1. Although the triggers are OOD data samples, the model is explicitly trained on them. To the trained model, these samples yield high-confidence predictions and low entropy, making them mathematically indistinguishable from "in-distribution" data. Standard OOD detectors (which typically rely on uncertainty or anomaly scores) would fail to flag them.
>
>     2.2. TraMark can utilize arbitrary synthetic data (e.g., WafflePattern) as triggers. The search space for such synthetic noise is effectively infinite. It is computationally infeasible for an attacker to "brute-force" or identify the specific trigger pattern used.
>
> > **[Q9] The repository link provided in the OpenReview abstract results in an error ("not found"). However, the link in the PDF appears to work.**
>
> **Response.** We sincerely apologize for the inconvenience caused by the broken link in the OpenReview abstract. We identified that the system incorrectly parsed the closing parenthesis as part of the URL. We have now corrected the link to ensure the repository is fully accessible.
>
> ---
>
> **We sincerely thank the reviewer again for your time and constructive comments. We hope these clarifications and additional results have fully resolved your concerns, and we remain open to any further discussion.**
>
> ---

---

### Author Response · Authors · 2025-11-21
**Official Comment by Authors**

Dear all reviewers,

We once again thank you for taking the time to read our paper carefully and for providing insightful comments and constructive suggestions that helped improve the quality of our work. We hope that our responses have addressed your concerns and questions. All revisions in the manuscript have been marked in **blue**.

*We are open to any further discussion.*

Regards,

Authors of submission 13024

---

### Author Response · Authors · 2025-12-03
**General Response to ACs and Senior ACs**

Dear Area Chairs and Senior Area Chairs,

We thank the Area Chairs for meta-reviewing our submission under the current special circumstances, and we sincerely appreciate the reviewers for their careful evaluation and constructive feedback.

---

Overall, the reviewers acknowledged the novelty and practicality of TraMark as a fully black-box, server-side traceable watermarking framework for federated learning, while raising questions mainly about detectability, scalability, system overhead, privacy, and advanced FL settings. We summarize how we addressed these concerns below.

---

Reviewer 1fMj raised concerns about watermark detectability, scalability to larger client populations, compatibility with robust aggregation, warm-up leakage, privacy assumptions, and OOD-based evasion. In response, we conducted a new statistical analysis on extreme parameter updates and showed that watermark parameters are indistinguishable from normal training updates, indicating the low detectability of our watermark. We added a new large-scale experiment on ImageNet-1k with 100 clients and provided a full system-level scalability analysis. We further introduced a perturbation-based defense against collusion with negligible utility loss, evaluated TraMark with a robust aggregation method, Multi-Krum, discussed early-leakage under different warm-up ratios, clarified the standard FL assumption, and addressed OOD-based evasion from both theoretical and practical perspectives.

---

Reviewer UwiB primarily questioned the computational overhead, communication cost, privacy–traceability trade-off, fairness across clients, and applicability to asynchronous FL. We clarified that the actual per-client injection cost of TraMark is only 0.67s, comparable to the state-of-the-art method FedTracker. We explained that communication overhead is unchanged in cross-silo FL since model distribution is already unicast. We added a detailed privacy–traceability discussion, reported per-client accuracies to verify fairness, and conducted new experiments integrating TraMark with FedAsync, showing full traceability under asynchronous updates.

---

Reviewer LhGb requested stronger ablation studies and a clearer evaluation of overhead. We added a new ablation comparing TraMark with and without parameter partitioning, showing that partitioning is essential to prevent both accuracy degradation and watermark collisions. We also revised the overhead evaluation to explicitly report per-client computation cost and updated all tables to report exact VR values instead of checkmarks.

---

Overall, we have substantially strengthened the manuscript by adding new large-scale experiments, new ablation studies, revised overhead evaluations, and expanded discussions on scalability, privacy, collusion, robust aggregation, and asynchronous FL. We believe that all major concerns raised by the reviewers have been fully addressed in the revised version. We hope this summary is helpful for the meta-review process.

Best regards,

Authors of Submission 13024

---

### Meta-Review · Area_Chair_iM2a · 2026-01-03

**Summary:**

The paper proposes TraMark, a black-box, server-side traceable watermarking framework for Federated Learning.
I recommend Accepting this paper. Although the initial reviews were mixed (Scores: 6, 4, 4), the authors provided an exceptionally strong rebuttal that addressed the primary concerns regarding scalability, detectability, and robustness.

**Reviewer Concerns:**

The authors must ensure that all the additional results provided in the rebuttal (especially the ImageNet experiments, the privacy-traceability discussion, and the overhead analysis) are fully incorporated into the final manuscript.

**Reviewer Scores:**

The initial borderline scores reflected concerns regarding experimental validation (e.g., scalability and overhead). Given that the rebuttal effectively addressed these issues with significant new experiments (e.g., ImageNet-1k), I believes the concerns are resolved and the paper now merits acceptance.

---

### Decision · Program_Chairs · 2026-01-26

Accept (Poster)